# Unraveling Feature Extraction Mechanisms in Neural Networks

**Xiaobing Sun , Jiaxi Li , Wei Lu**
StatNLP Research Group
Singapore University of Technology and Design
{xiaobing_sun, jiaxi_li}@mymail.sutd.edu.sg, wei_lu@sutd.edu.sg

## Abstract

The underlying mechanism of neural networks in capturing precise knowledge has been the subject of consistent research efforts. In this work, we propose a theoretical approach based on Neural Tangent Kernels (NTKs) to investigate such mechanisms. Specifically, considering the infinite network width, we hypothesize the learning dynamics of target models may intuitively unravel the features they acquire from training data, deepening our insights into their internal mechanisms. We apply our approach to several fundamental models and reveal how these models leverage statistical features during gradient descent and how they are integrated into final decisions. We also discovered that the choice of activation function can affect feature extraction. For instance, the use of the *ReLU* activation function could potentially introduce a bias in features, providing a plausible explanation for its replacement with alternative functions in recent pre-trained language models. Additionally, we find that while self-attention and CNN models may exhibit limitations in learning n-grams, multiplication-based models seem to excel in this area. We verify these theoretical findings through experiments and find that they can be applied to analyze language modeling tasks, which can be regarded as a special variant of classification. Our contributions offer insights into the roles and capacities of fundamental components within large language models, thereby aiding the broader understanding of these complex systems.

## 1 Introduction

Neural networks have become indispensable across a variety of natural language processing (NLP) tasks. There has been growing interest in understanding their successes and interpreting their characteristics. One line of works attempts to identify possible features captured by them for NLP tasks (Li et al., 2016; Linzen et al., 2016; Jacovi et al., 2018; Hewitt and Manning, 2019; Vulić et al., 2020). They mainly develop empirical methods to verify hypotheses regarding the semantic and syntactic features encoded in the output. Such works

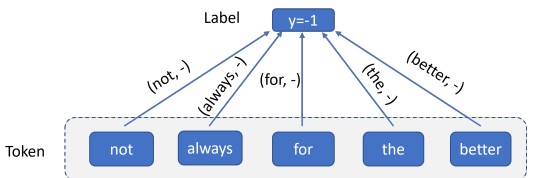

Figure 1: Example of co-occurrence features between tokens and labels (self-attention model).

may result in interesting findings, but those models still remain *black-boxes* to us. Another line seeks to reveal internal mechanisms of neural models using mathematical tools (Levy and Goldberg, 2014; Saxe et al., 2013; Arora et al., 2018; Bhojanapalli et al., 2020; Merrill et al., 2020; Dong et al., 2021; Tian et al., 2023), which can be more straightforward and insightful. However, few of them have specifically focused on the feature extraction of neural NLP models.

When applying neural models to downstream NLP tasks in practice, we often notice some modules perform better than others on specific tasks, while some exhibit similar behaviors. We may wonder what mechanisms are behind such differences and similarities between those modules. By acquiring deeper insights into the roles of those modules in a complex model with respect to feature extraction, we will be able to select or even design more suitable models for downstream tasks.

In this work, we propose a novel theoretical approach to understanding the mechanisms, through which fundamental models (often used as modules in complex models) acquire features during gradient descent in text classification tasks. The evolution of model output can be described as learning dynamics involving NTKs (Jacot et al., 2018; Arora et al., 2019), which are typically used to study various properties of neural networks, including convergence and generalization. While these representations can be complex in practice, when the width of the network approaches infinity, they tend to converge to less complex representations and remain asymptotically constant (Jacot et al., 2018), allowing us to intuitively interpret the learning dynamics and identify the relevant features cap-

tured by the model.

We applied our approach to several fundamental models, including a multi-layer perceptron (MLP), a convolutional neural network (CNN), a linear Recurrent Neural Network (L-RNN), a self-attention (SA) model (Vaswani et al., 2017), and a matrix-vector (MV) model (Mitchell and Lapata, 2009) and exhibit the MLP, CNN, and SA models may behave similarly in capturing token-label features, while the MV and L-RNN extract different types of features. Our contributions include:

- We propose an approach to theoretically investigate feature extraction mechanisms for fundamental neural models.

- We identify significant factors such as the choice of activation and unveil the limitations of these models, e.g., both the CNN and SA models may not effectively capture meaningful n-gram information beyond individual tokens.

- Our experiments validate the theoretical findings and reveal their relevance to advanced architectures such as Transformers (Vaswani et al., 2017).

Our intention through this work is to provide new insights into the core components of complex models. By doing so, we aim to contribute to the understanding of the behaviors exhibited by state-of-the-art large language models and facilitate the development of enhanced model designs[1].

## 2 Related Work

**Probing features for NLP models** Probing linguistic features is an important topic for verifying the interpretability of neural NLP models. Li et al. (2016) employed a visualization approach to detect linguistic features such as negation captured by the hidden states of LSTMs. Linzen et al. (2016) examined the ability of LSTMs to capture syntactic knowledge using number agreement in English subject-verb dependencies. Jacovi et al. (2018) studied whether the CNN models could capture n-gram features. Vulić et al. (2020) presented a systematic analysis to probe possible knowledge that the pre-trained language models could implicitly capture. Chen et al. (2020) proposed an algorithm to detect hierarchical feature interaction for text classifiers. Empirically, such work reveals that neural NLP models can capture useful and interpretable features for downstream tasks. Our work seeks to explain how neural NLP models capture useful features during training from a theoretical perspective.

**Infinite-width Neural Networks** Researchers found that there could be interesting patterns when the neural network's width approaches infinity. Lee et al. (2018) linked infinitely wide deep networks to Gaussian Processes. A recent line of work (Jacot et al., 2018; Bietti and Mairal, 2019; Nguyen et al., 2021; Loo et al., 2022) proposed that as the network width approaches infinity, the dynamics can be characterized by the NTK, which converges to a kernel determined at initialization and remains constant. This conclusion holds for fully-connected neural networks, CNNs (Arora et al., 2019) and RNNs (Emami et al., 2021; Alemohammad et al., 2021). Later, Yang and Littwin (2021) showed that such properties of NTKs can be applied to a randomly initialized neural network of any architecture. Very limited studies have delved into the analysis of feature extraction in neural NLP models. We will investigate the internal mechanisms of neural NLP models under extreme conditions.

## 3 Analysis

We use learning dynamics to describe the updates of neural models during training with the aim of identifying potentially useful properties. For the ease of presentation and discussion, we focus on binary text classification[2].

**Model Description** Assume we have a training dataset denoted by $\mathcal{D}$, consisting of $m$ labeled instances. Let $\mathcal{X}$ and $\mathcal{Y}$ represent all the sentences and labels in the training dataset, respectively. $x \in \mathcal{X}$ is an instance consisting of a sequence of tokens, and $y \in \mathcal{Y}$ is the corresponding label. The vocabulary size is $|V|$. Consider a binary text classification model, where $y \in \{-1, +1\}$. The model output, denoted as $s(t) \in \mathbb{R}$ at time $t$ is

$$s(t) = \boldsymbol{f}_t(x; \boldsymbol{\theta}_t), \qquad (1)$$

where $\boldsymbol{\theta}_t$ (a vector) is the concatenation of all the parameters, which are functions of time $t$. We refer to the model output $s(t)$ as the *label score* at time $t$. This score is used for classification decisions, *positive* if $s(t) > 0$ and *negative* otherwise.

**Learning Dynamics** The evolution of a label score can be described by learning dynamics, which may indicate interesting properties. Let $\boldsymbol{f}_t(\mathcal{X}) \in \mathbb{R}^m$ represent the concatenation of all the outputs of training instances at time $t$, and $y \in \mathcal{Y}$

---

[1]Our code is available at https://github.com/richardsun-voyager/ufemnn.

[2]Analysis for multi-class classification can be found in Appendix A.

is the desired label. Given a test input $x'$, the corresponding label score $s'(t)$ follows the dynamics

$$\begin{aligned} \dot{s}'(t) &= \nabla_\theta f_t^\top(x') \nabla_{\theta_t} \boldsymbol{f}_t(\mathcal{X}) \nabla_{\boldsymbol{f}_t(\mathcal{X})} \mathcal{L} \\ &= \Theta_t(x', \mathcal{X}) \nabla_{\boldsymbol{f}_t(\mathcal{X})} \mathcal{L}, \end{aligned} \quad (2)$$

where $\Theta_t(x', \mathcal{X})$ is the NTK at time $t$ and $\mathcal{L}$ is the empirical loss defined as

$$\mathcal{L} = -\frac{1}{m} \sum_{(x,y) \in \mathcal{D}} \log g(ys). \quad (3)$$

where $g$ is the *sigmoid* function. For simplicity, we will omit the time stamp $t$ in our subsequent notations. The dynamics $\dot{s}'$ will obey

$$\dot{s}' = \frac{1}{m} \sum_{(x,y) \in \mathcal{D}} g(-ys^{(x)}) y \Theta(x', x), \quad (4)$$

where $s^{(x)}$ is the label score for the training instance $x$. Obtaining closed-form solutions for the differential equation in Equation 4 is a challenge. We thereby consider an extreme scenario with the infinite network width, suggested by Lee et al. (2018).

**Infinite-Width** When the network width approaches infinity, the NTK will converge and stay constant during training (Jacot et al., 2018; Arora et al., 2019; Yang and Littwin, 2021). Therefore, the learning dynamics can be written as follows,

$$\dot{s}' = \frac{1}{m} \sum_{(x,y) \in \mathcal{D}} g(-ys^{(x)}) y \Theta_\infty(x', x), \quad (5)$$

where $\Theta_\infty(x', x)$ refers to the converged NTK determined at initialization. This convergence may allow us to simplify the representations of the learning dynamics and offer more intuitive insights to analyze its evolution over time.

There can be certain interesting properties (regarding the trend of the label scores) harnessed by the interaction $y\Theta_\infty(x', x)$, where $y$ controls the direction and $\Theta_\infty(x', x)$ may indicate the relationship between $x'$ and $x$. Certain hypotheses can be drawn from these properties. First, the converged NTK $\Theta_\infty(x', x)$ may intuitively represent the interaction between the test input $x'$ and the training instance $x$. This could extend to the interaction between the basic units (tokens or n-grams) from $x'$ and $x$, as the semantic meaning of an instance can be deconstructed into the combination of the meanings of its basic units (Mitchell and Lapata, 2008; Socher et al., 2012). Second, if $\Theta_\infty(x', x)$ depends on the similarity between $x'$ and $x$, a more

deterministic trend can be predicted for a test input $x'$ that closely resembles the training instances of a specific type. For example, suppose $\Theta_\infty(x', x)$ exhibits a significantly large gain when $x'$ is similar to $x$ at a particular $y$, and the dynamics will likely receive significant gains in a desired direction during training, thus enabling us to predict the trend of the label score.

We thereby propose the following approach to investigate a target model and verify our aforementioned hypotheses: 1) redefining the target model following the settings proposed by Jacot et al. (2018); Yang and Littwin (2021), which guarantees the convergence of NTKs; 2) obtaining the converged NTK $\Theta_\infty(x', x)$ and the learning dynamics under the infinite-width condition; 3) performing analysis on the learning dynamics of basic units and revealing possible features.

## 4 Interpreting Fundamental Models

We investigate an MLP model, a CNN model, an SA model, an MV model, and an L-RNN model, respectively. Details and proofs for the lemmas and theorems can be found in Appendix A.

**Notation** Let $\boldsymbol{e} \in \mathbb{R}^{|V|}$ be the one-hot vector for token $e$, $l^{(x)}$ be the instance length, $\boldsymbol{W}^e \in \mathbb{R}^{d_{in} \times |V|}$ be the weight of the embedding layer, and $\boldsymbol{v} \in \mathbb{R}^{d_{out}}$ be the final layer weight. $\boldsymbol{W} \in \mathbb{R}^{d_{out} \times d_{in}}$ is the weight of the hidden layer in the MLP model. $\boldsymbol{W}_k^c \in \mathbb{R}^{d_{out} \times d_{in}}$ is the kernel weight corresponding to the $k$-th token in the sliding window in the CNN model. For simplicity, we let $d_{out} = d_{in} = d$. Assume all the parameters are initialized with Gaussian distributions in our subsequent analysis, i.e., $\boldsymbol{W}_{ij} \sim \mathcal{N}(0, \sigma_w^2)$, $\boldsymbol{W}_{ij}^e \sim \mathcal{N}(0, \sigma_e^2)$, and $\boldsymbol{v}_j \sim \mathcal{N}(0, \sigma_v^2)$, and $\boldsymbol{W}_{ij}^c \sim \mathcal{N}(0, \sigma_w^2)$, for the sake of NTK convergence.

### 4.1 MLP

Following Wiegreffe and Pinter (2019), given instance $x$, the output of MLP is defined as

$$s = \frac{\boldsymbol{v}^\top}{\sqrt{d}} \sum_{j=1}^{l^{(x)}} \boldsymbol{\phi}(\boldsymbol{W} \frac{1}{\sqrt{d}} \boldsymbol{W}^e \boldsymbol{e}_j). \quad (6)$$

The label score $s$ will be used for making classification decisions. $\phi$ is the element-wise *ReLU* function. $\boldsymbol{e}_j$ is the *one-hot* vector for token $e_j$. It is not straightforward to analyze $s$ directly, which can be viewed as the sum of token-level label scores. Instead, as basic units are tokens in this model, we focus on the label score of every single token and understand how they contribute to the instance-level label score. When the test input $x'$ is simply

a token $e$, we can get the corresponding NTK with the infinite network width.

**Lemma 4.1.** When $d \to \infty$, the NTK between the token $e$ and instance $x$ in the MLP model converges to

$$\Theta_\infty(e, x) = \rho \sum_{j=1}^{l(x)} e^\top e_j + \sum_{j=1}^{l(x)} \mu, \quad (7)$$

where $\rho = \frac{(\pi-1)\sigma_e^2 \sigma_w^2}{2\pi} + \frac{\sigma_e^2 \sigma_v^2 + \sigma_w^2 \sigma_v^2}{2}$ and $\mu = \frac{\sigma_e^2 \sigma_w^2}{2\pi}$.

Note that, for two tokens $e_j$ and $e_k$, their one-hot vectors satisfy $e_j^\top e_k = 0$ if $e_j \neq e_k$; $e_j^\top e_k = 1$ if $e_j = e_k$. The dot-product $\sum_{j=1}^{l(x)} e^\top e_j$ can be interpreted as the frequency of $e$ appearing in instance $x$.

**Theorem 4.2.** The learning dynamics of token $e$'s label score obey

$$\dot{s}^e = \frac{\rho}{m} \sum_{(x,y)\in\mathcal{D}} g(-ys^{(x)}) y \omega(e, x)$$
$$+ \frac{\mu}{m} \sum_{(x,y)\in\mathcal{D}} g(-ys^{(x)}) y l^{(x)}, \quad (8)$$

where $\omega(e, x) = \sum_{j=1}^{l(x)} e^\top e_j$, which depends on the training data and will not change over time.

The *non-linearity* of the sigmoid function $g(-ys)$ makes it a challenge to obtain a *closed-form* solution for the dynamics.

However, we can predict trends for the label scores in special cases. Note that the polarity of the first term in Equation 8 will depend on $y\omega(e, x)$ in each training instance. For instance, consider a token that only appears in positive instances, i.e., $\omega(e, x) > 0$ when $y = +1$; $\omega(e, x) = 0$ when $y = -1$. In this case, the first term remains positive and incrementally contributes to the label score $s^e$ throughout the training process. The opposite trend occurs for tokens solely appearing in negative instances. If the impact of the second term is minimal, the label scores of these two types of tokens will be significantly positive or negative after sufficient updates. The final classification decisions are made based on the linear combination of the label scores for the constituent tokens. The second term in Equation 8 is unaffected by $\omega(e, x)$ and is shared by all the tokens $e$ at each update. It can be interpreted as an induced feature bias. Particularly, when this term is sufficiently large, it may cause an imbalance between the tokens co-occurring with the positive label and those co-occurring with the negative label, rendering one type of tokens more influential than the other for classification.

Theorem 4.2 may explain how the MLP model leverages the statistical co-occurrence features between $e$ and $y$ as shown in Figure 1, and integrate them in final classification decisions, i.e., tokens solely appearing in positive/negative instances will likely contribute in the direction of predicting a positive/negative label.

## 4.2 CNN

We consider the 1-dimensional CNN, with kernel size, stride size, and padding size set to $K$, 1, and $K - 1$ respectively. For each sliding window $c_j$ comprising $K$ consecutive tokens, the corresponding feature $c_j \in \mathbb{R}^d$ can be represented as

$$c_j = \sum_{k=1}^{K} W_k^c \frac{1}{\sqrt{d}} W^e e_{j+k-1}, \quad (9)$$

where $W_k^c$ is the kernel weight corresponding to the $k$-th token in the sliding window.

The label score of an instance is computed as

$$s = \frac{v^\top}{\sqrt{d}} \sum_{j=-(K-1)}^{l(x)} \phi(c_j), \quad (10)$$

where $-(K - 1)$ means the position for the left-most padding token. The first and last $K - 1$ padding tokens in an instance are represented by zero vectors. $\phi$ is the element-wise *ReLU* function. For brevity, we will denote $\sum_{j=-(K-1)}^{l(x)}$ by $\sum_j$.

Let us focus on a single sliding window and study the learning dynamics of its label score.

**Lemma 4.3.** Consider a sliding window $c$ consisting of tokens $e_1, e_2, \ldots, e_K$, when $d \to \infty$ the NTK between $c$ and instance $x$ converges to

$$\Theta_\infty(c, x) = \sum_j F[\omega_c(c, c_j)] +$$
$$\rho \sum_{k=1}^{K} \sum_j H[\omega_c(c, c_j)] e_k^\top e_{j+k-1}, \quad (11)$$

where
$$\omega_c(c, c_j) = \sum_{k'=1}^{K} e_{k'}^\top \sum_{k=1}^{K} e_{j+k-1}, \quad \rho = \sigma_v^2(\sigma_e^2 + \sigma_w^2).$$

$\omega_c$ means the number of shared tokens between $c$ and $c_j$ regardless of positions. $F$ and $H$[3] are *monotonically-increasing* and *non-negative* functions depending on $\sigma_e^2 \sigma_w^2$.

The first term in $\Theta_\infty(c, x)$ captures the token similarity between sliding windows $c$ and $c_j$ regardless of token positions. In the second term, $\sum_j H[\omega_c(c, c_j)] e_k^\top e_{j+k-1}$ can be viewed as the weighted frequency of token $e_k$ in instance $x$, and when $\sigma_v$ is sufficiently large, the converged NTK is majorly influenced by the sum of the weighted frequencies of the tokens in $c$ appearing in $x$.

---

[3]Their definitions can be found in Appendix A.2.

**Theorem 4.4.** The dynamics of the label score of the test sliding window $c$ obey

$$\dot{s}^c = \frac{\rho}{m} \sum_{k=1}^{K} \sum_{(x,y)\in\mathcal{D}} g(-ys^{(x)}) y\omega(e_k, x)$$
$$+ \frac{1}{m} \sum_{(x,y)\in\mathcal{D}} g(-ys^{(x)}) y \sum_j F[\omega_c(c, c_j)],$$

$$(12)$$

where $\omega(e_k, x) = \sum_j H[\omega_c(c, c_j)] e_k^\top e_{j+k-1}$.

Theorem 4.4 indicates that with a sufficiently large $\sigma_v$, the learning dynamics for window $c$ may mainly depend on the linear combination of the weighted learning dynamics of its constituent tokens. Similar analysis can be performed on the label score of the sliding window. This may not exactly encode n-grams, which are inherently sensitive to order and can extend beyond their constituent elements. Instead, for each window, it is more akin to the composition model based on vector addition as described in the work of Mitchell and Lapata (2009). The second term in Equation 12 may not be zero even if $c$ shares no tokens with $x$, suggesting there can be an induced feature bias similar to the one in the MLP model.

When $c$ only shares tokens with either positive or negative instances, regardless of position, the corresponding label score will receive relatively large gains in one direction during updates. This means the CNN model also captures co-occurrence features between tokens and labels. Importantly, a single token can also be viewed as a sliding window, padded with additional tokens, thereby leading to conclusions about the trend of label scores that mirror those drawn from the MLP model.

### 4.3 SA

We employ a fundamental self-attention module, analogous to the component found in Transformers. The representation of the $i$-th output in the instance will be computed as a weighted sum of token representations as follows,

$$h_i = \sum_{j=1}^{l^{(x)}} \frac{\alpha_{ij}}{\sqrt{d}} W^e e_j,$$

$$(13)$$

where $\alpha_{ij}$ is the weight produced by a *softmax* function as follows,

$$\alpha_{ij} = \frac{\exp(a_{ij})}{\sum_{j'=1}^{l^{(x)}} \exp(a_{ij'})}.$$

$$(14)$$

We define the attention score $a_{ij}$ from position $i$ to $j$ as

$$a_{ij} = \frac{(W^e e_i + P_i)^\top (W^e e_j + P_j)}{d},$$

$$(15)$$

where $P_i$ ($P_j$) is the positional embedding at position $i$ ($j$) and will be fixed during training. The instance label score will be computed as

$$s = v^\top \sum_{i=1}^{l^{(x)}} h_i = \sum_{i=1}^{l^{(x)}} \sum_{j=1}^{l^{(x)}} \frac{\alpha_{ij}}{\sqrt{d}} v^\top W^e e_j, \quad (16)$$

which can be viewed as the weighted sum of token-level label scores if we define such a score for each token $e$ as $s_e = \frac{1}{\sqrt{d}} v^\top W^e e$. We consider the case where the test input is also simply a token $e$.

**Lemma 4.5.** When $d \to \infty$, the NTK between the token $e$ and the instance $x$ will converge to $\Theta_\infty(e, x)$, which obeys

$$\Theta_\infty(e, x) \approx (\sigma_e^2 + \sigma_v^2) \sum_{i=1}^{l^{(x)}} \sum_{j=1}^{l^{(x)}} \mathbb{E}(\alpha_{ij}) e^\top e_j,$$

$$(17)$$

where $\mathbb{E}(\alpha_{ij})$ is the expectation of $\alpha_{ij}$.

**Theorem 4.6.** The learning dynamics of the label score of a token $e$ obey

$$\dot{s}^e = \frac{\rho}{m} \sum_{(x,y)\in\mathcal{D}} g(-ys^{(x)}) y\omega(e, x),$$

$$(18)$$

where $\omega(e, x) = \sum_{i=1}^{l^{(x)}} \sum_{j=1}^{l^{(x)}} \mathbb{E}(\alpha_{ij}) e^\top e_j$ and $\rho = \sigma_e^2 + \sigma_v^2$.

Theorem 4.6 shows the learning dynamics of token e's label score also depends on the weighted sum of the frequencies of $e$ appearing in $x$. The learning dynamics of a single token's label score will likely resemble that in the MLP model, capturing the co-occurrence features between tokens and labels despite the weights. This model may not experience an induced bias, compared to the MLP model as discussed in Theorem 4.2. This will be further explored in our experiments.

### 4.4 MV

We consider the matrix-vector representation as applied in adjective-noun composition (Baroni and Zamparelli, 2010) and recursive neural networks (Socher et al., 2012). It models each word pair through matrix-vector multiplication. The label score of an instance is defined as

$$s = v^\top \sum_j \frac{1}{d\sqrt{d}} M(e_j) W^e e_{j+1},$$

$$(19)$$

where $M(e_j) = \text{diag}(WW^e e_j)$ (diag converts a vector into a diagonal matrix.) and $j = 1, 2, \ldots, l^{(x)} - 1$.

**Lemma 4.7.** Given a bigram consisting of two tokens $e_a e_b$, with the infinite network width the NTK will converge to

| Model | Feature | Bias | Activation | Definition | NTK |
|---|---|---|---|---|---|
| MLP | token-label | Yes | ReLU | $s = \frac{\boldsymbol{v}^\top}{\sqrt{d}} \sum_{j=1}^{l(x)} \phi(\boldsymbol{W} \frac{1}{\sqrt{d}} \boldsymbol{W}^e \boldsymbol{e}_j)$ | $\Theta_\infty(e,x) = \rho \sum_{j=1}^{l(x)} \boldsymbol{e}^\top \boldsymbol{e}_j + \sum_{j=1}^{l(x)} \mu$ |
| CNN | token-label | Yes | ReLU | $s = \frac{\boldsymbol{v}^\top}{\sqrt{d}} \sum_{j=-(K-1)}^{l(x)} \phi(\boldsymbol{c}_j)$ | $\Theta_\infty(c,x) = \sum_j F[\omega_c(c,c_j)] + \rho \sum_{k=1}^{K} \sum_j H[\omega_c(c,c_j)] \boldsymbol{e}_k^\top \boldsymbol{e}_{j+k-1}$ |
| SA | token-label | No | - | $s_i = \frac{\boldsymbol{v}^\top}{\sqrt{d}} \sum_{j=1}^{l(x)} \alpha_{ij} \boldsymbol{W}^e \boldsymbol{e}_j$ | $\Theta_\infty(e,x) \approx \rho \sum_{i=1}^{l(x)} \sum_{j=1}^{l(x)} \mathbb{E}(\alpha_{ij}) \boldsymbol{e}^\top \boldsymbol{e}_j$ |
| MV | bigram-label | No | - | $s = \boldsymbol{v}^\top \sum_j \frac{1}{d\sqrt{d}} \boldsymbol{M}(\boldsymbol{e}_j) \boldsymbol{W}^e \boldsymbol{e}_{j+1}$ | $\Theta_\infty(e_a e_b, x) = \rho \sum_j \boldsymbol{e}_j^\top \boldsymbol{e}_a \boldsymbol{e}_{j+1}^\top \boldsymbol{e}_b$ |
| L-RNN | token-label-position | - | - | $s = \frac{1}{d} \sum_{j=1}^{T} \boldsymbol{v}^\top (\frac{\boldsymbol{W}^h}{\sqrt{d}})^{T-j} \boldsymbol{W} \boldsymbol{W}^e \boldsymbol{e}_j$ | $\Theta_\infty(e,k,x) = \rho(k) \boldsymbol{e}^\top \boldsymbol{e}_{l(x)-k}$ |

Table 1: Co-occurrence features captured by target models. "$k$" in L-RNN refers to the distance from the last tokens. $\rho$ ($\mu$) refers to the non-negative coefficient determined at initialization for each model. $e$ and $x$ refer to a token and an instance, respectively. "-" means *not applicable*.

$$\Theta_\infty(e_a e_b, x) = (\sigma_e^2 + 3\sigma_v^2)\sigma_e^2 \sigma_w^2 \sum_j \boldsymbol{e}_j^\top \boldsymbol{e}_a \boldsymbol{e}_{j+1}^\top \boldsymbol{e}_b. \tag{20}$$

It is worth highlighting that the interaction $\boldsymbol{e}_j^\top \boldsymbol{e}_a \boldsymbol{e}_{j+1}^\top \boldsymbol{e}_b$ is different from the interaction resulting from the aforementioned models. When $e_a \equiv e_j$ and $e_b \equiv e_{j+1}$ (i.e., $e_a e_b \equiv e_j e_{j+1}$), the NTK will gain a relatively large value, implying the ability to capture co-occurrence knowledge between bigrams and labels.

**Theorem 4.8.** The dynamics of the label score of the test bigram $e_a e_b$ obey

$$\dot{s}^{ab} = \frac{\rho}{m} \sum_{(x,y)\in\mathcal{D}} g(-ys^{(x)}) y \omega(e_a e_b, x), \tag{21}$$

where $\rho = (\sigma_e^2 + 3\sigma_v^2)\sigma_e^2 \sigma_w^2$ and $\omega(e_a e_b, x) = \sum_j \boldsymbol{e}_j^\top \boldsymbol{e}_a \boldsymbol{e}_{j+1}^\top \boldsymbol{e}_b$.

Here, $\omega(e_a e_b, x)$ can be viewed as the frequency of bigram $e_a e_b$ seen in instance $x$. Specifically, when a bigram co-occurs with a positive (negative) label, it will receive a positive (negative) gain during gradient descent.

We provide the analysis of the L-RNN model in Appendix A. The features captured by different architectures are listed in Table 1.

## 5 Experiments

We conduct experiments to verify our aforementioned analysis in the following aspects: a) verify the features acquired by our models; b) explore factors that may affect feature extraction; c) examine the limitations of those models.

**Datasets** We consider the following datasets: **SST**, instances with "positive" and "negative" labels are extracted from the original Stanford Sentiment Treebank dataset (Socher et al., 2013). We also extract instances with sub-phrases (along with labels) under the name "SSTwsub". **Agnews**, the AG-news dataset, which consists of titles and description fields of news articles from 4 classes, "World", "Sports", "Business" and "Sci/Tech". **IMDB**, the binary IMDB dataset (Maas et al., 2011)

| Data | Train | Valid | Test | |V| | Len |
|---|---|---|---|---|---|
| SST | 6,920 | 872 | 1,821 | 16,174 | 18 |
| SSTwsub | 98,794 | 872 | 1,821 | 17,404 | 8 |
| IMDB | 39,877 | 5,016 | 5,107 | 146,582 | 270 |
| Agnews | 110,000 | 10,000 | 7,600 | 85,568 | 36 |

Table 2: Dataset statistics. "Train", "Valid", and "Test" refer to the training, validation, and test sets, respectively. "|V|" refers to the vocabulary size and "Len" refers to the average training instance length.

which consists of movie reviews with relatively longer texts. The statistics are listed in Table 2.

In addition, Penn Tree Bank (PTB) (Marcus et al., 1993), WIKITEXT2 (Wiki2), and a Shakespeare dataset are considered for language modeling, a special classification variant[4].

**Setup** We randomly initialize all the parameters with Gaussian distributions. Unless specified otherwise, the variances of the parameters are set as 0.01. While our analysis is based on vanilla gradient descent, training models using SGD optimizers with a small learning rate can be challenging. Therefore, Adagrad (Duchi et al., 2011) optimizers[5] are used in practice. The network width $d$ is set as 64. To verify the features learned by the models, we extract corresponding co-occurrence pairs for each model from training data. Specifically, for the MLP, CNN, and SA models, we calculate the *token-label* frequencies from the training data. For example, if a token $e$ co-occurs three times more frequently with the positive label (+) than with the negative label (-), i.e., $\frac{freq(e,+)}{freq(e,-)} \geq 3$, we will extract this *(e, +)* pair[6]. For the MV and L-RNN models, we calculate *bigram-label* and *token-label-position* frequencies respectively, and extract co-occurrence pair in a similar way. For simplicity, tokens (bigrams) co-occurring with the positive/negative label will be referred to as *positive/negative tokens (bigrams)*.

---

[4]More details are listed in Appendix C.
[5]The Adam (Kingma and Ba, 2014) optimizer is also considered in Appendix C.
[6]The conditions in the theoretical analysis are relaxed in experiments.

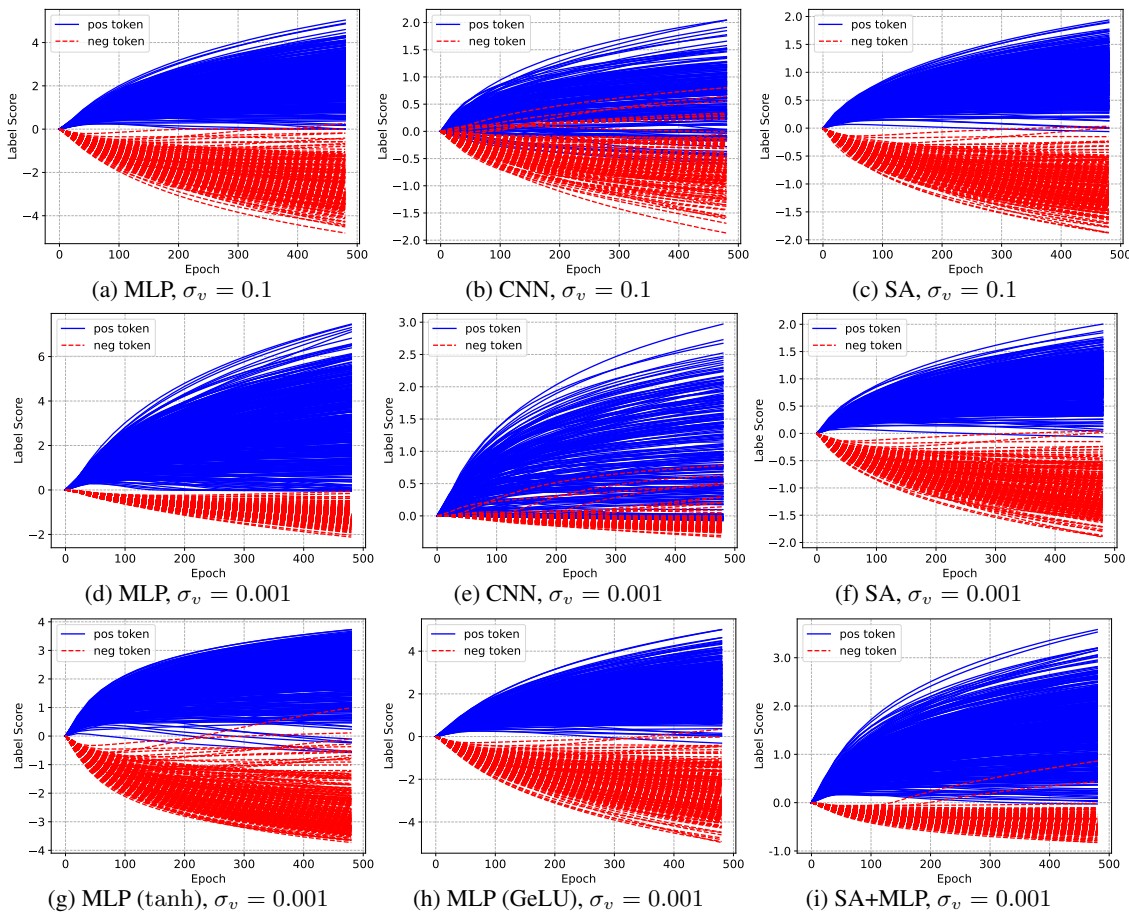

Figure 2: Evolution of the label scores for the extracted tokens from SST over epochs. "pos token" and "neg token" refer to "positive tokens" and "negative tokens" respectively.

## 5.1 Feature Extraction

We illustrate the label scores for the extracted co-occurrence pairs to examine the features predicted by our approach. It can be seen from Figures 2a, 2b, and 2c that the label scores for tokens in the extracted co-occurring pairs evolve as expected over epochs for the MLP, CNN, and SA models. The label scores of the tokens co-occurring majorly with the positive label consistently receive positive gains during training, whereas those of the tokens co-occurring majorly with the negative label experience negative gains, thus playing opposite roles in final classification decisions. Similar patterns can be observed on IMDB in Appendix C. We also extract bigrams co-occurring majorly with either the positive or negative label from SSTwsub and calculate their label scores using a trained MV model, which exhibits the capability of capturing the co-occurrence between bigrams and labels as shown in Figure 3.

Our analysis on the binary classification tasks can be extended to the multi-class classification scenario on Agnews of four-class. The label scores for the tokens associated with a specific class would

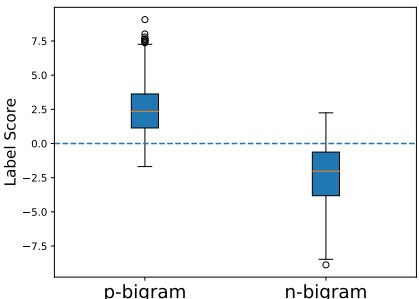

Figure 3: Distribution of the label scores for extracted bigrams from SSTwsub. "p" refers to *positive* and "n" refers to *negative*.

be assigned relatively large scores in the dimension corresponding to the class as shown in Figures 4a, 4b, 4c, and 4d. These observations support our analysis of the feature extraction mechanisms within our target models.

In addition, we extend our experiments to language modeling tasks, which can be viewed as a variant of multi-class classification tasks, with the label space equivalent to the vocabulary size. Interestingly, we observe similar token-label patterns on Transformer-based language models incorporating self-attention modules despite their complexity, in

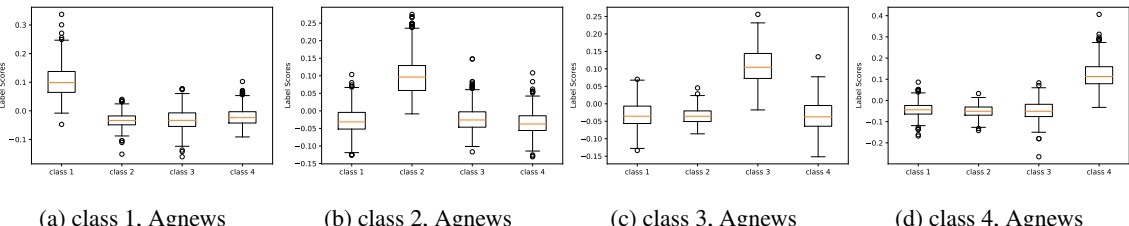

| (a) class 1, Agnews | (b) class 2, Agnews | (c) class 3, Agnews | (d) class 4, Agnews |

Figure 4: Label scores for extracted tokens from Agnews, a dataset with four classes. SA model. $d = 64$.

both word and character levels. Particularly, we find that nanoGPT, a light-weight implementation of GPT, can capture the co-occurrence features between context characters and target characters on the character-level Shakespeare dataset, and reflect them in the label scores as shown in Figure 5. Given a context character, the model's output is more likely to assign higher scores to target characters that predominantly co-occur with this context character in the training data, thereby making those target characters more likely to be predicted. This implies the significance of a large dataset may be (partially) ascribed to rich co-occurrence information between tokens. Further details can be found in Appendix C.

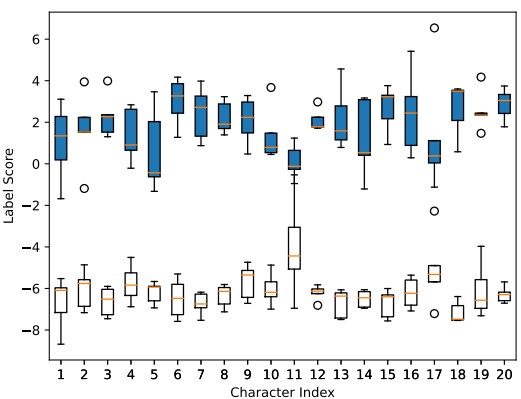

Figure 5: Distribution of the label scores for target characters majorly (blue) and rarely co-occurring with each extracted context character. nanoGPT. Shakespeare Dataset.

**Induced Bias**  Our approach also indicates that factors such as activation and initial weight variances could affect feature extraction. We downscale the variances for the final layer weight vectors at initialization and compare the learning curves of the extracted tokens' label scores from models with different activations. As can be seen from Figures 2d and 2e, a smaller initialization of the final layer weight variance can lead to a large feature bias, rendering negative tokens less significant than positive ones in the MLP and CNN models. This may not be a desirable situation, as Table 3 suggests a performance decline for the MLP model with *ReLU*. Furthermore, we compare other acti-

| Dataset | | **ReLU** | | **tanh** | | **GeLU** | | **SiLU** | |
|---|---|---|---|---|---|---|---|---|---|
| | | I | II | I | II | I | II | I | II |
| SST | valid | 78.4 | 68.0 | 77.2 | 77.3 | 78.0 | 78.3 | 77.8 | 77.8 |
| | test | 80.0 | 67.3 | 78.9 | 78.9 | 79.9 | 79.8 | 79.7 | 79.9 |
| Agnews | valid | 91.1 | 90.4 | 91.1 | 90.7 | 90.8 | 90.0 | 90.7 | 90.1 |
| | test | 91.0 | 90.2 | 90.6 | 90.1 | 90.6 | 89.6 | 90.6 | 89.6 |
| IMDB | valid | 89.5 | 89.6 | 89.8 | 89.7 | 91.5 | 91.7 | 91.5 | 91.8 |
| | test | 89.9 | 89.5 | 89.9 | 90.2 | 91.2 | 91.4 | 91.2 | 91.4 |

Table 3: Average accuracy (%) on SST with scaled variances (I: $\sigma_v = 0.1$ and II: $\sigma_v = 0.001$) and different activation functions. 3 trials for each run. MLP model.

vation functions such as $\tanh$, *GeLU*, and *SiLU*, which are alternatives to *ReLU*[7]. Figures 2g and 2h show that these alternatives are more robust than *ReLU* in the MLP model. This also suggests that while non-linear activations may not significantly alter the nature of learned features during training, they can affect the balance of the extracted features. Figure 2f shows the SA model is also robust to the change in initialization. However, incorporating an MLP with *ReLU* activation after the SA model reintroduces bias, as can be observed in Figure 2i, suggesting a possible reason why *ReLU* was replaced in models such as BERT (Devlin et al., 2019), GPT3 (Brown et al., 2020), and LLaMA (Touvron et al., 2023), despite its presence in the original Transformer architecture (Vaswani et al., 2017).

**Models' Limitations**  We aim to examine whether the CNN and SA models have a limitation in encoding n-grams in situations beyond constituent tokens' semantic meanings. We choose negation phenomena as our testbed, where a negation token can (partially) reverse the meanings of both positive and negative phrases, a task that is challenging to achieve by linear combination. We run experiments on the SSTwsub dataset with labeled sub-phrases, which contains rich negation phenomena, i.e., phrases with their negation expressions achieved by prepending negation tokens such as *not* and *never*. We extract positive and negative adjectives and create their corresponding negation expressions by prepending the negation word *not*. Figure 6a shows that the SA model can

---

[7]The discussion of $\tanh$ and visualization of *SiLU* can be found in Appendix B and Appendix C, respectively.

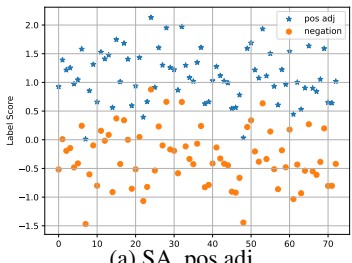 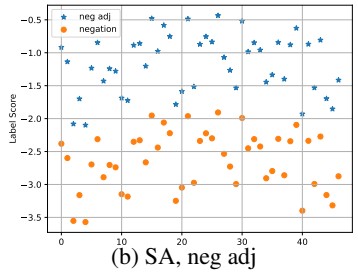 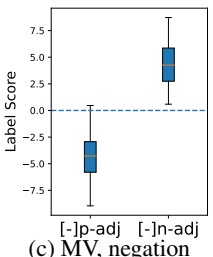

(a) SA, pos adj       (b) SA, neg adj       (c) MV, negation

Figure 6: Label scores for the extracted *positive adjectives* (pos/p adj) and *negative adjectives* (neg/n adj), as well as their negation expressions. SSTwsub. "[-]" refers to the negation operation.

capture negation phenomena for positive adjectives but does not perform well for negative adjectives as shown in Figure 6b. Specifically, prepending a negation word to the negative adjectives does not alleviate their negativity as expected but leads to the contrary. Based on our analysis, the polarity of a negation expression relies largely on the linear combination of the tokens' polarity in the SA model. As both the negation word *not*[8] and negative adjectives are assigned negative scores, their linear combination will still be negative. This is not a desirable case and not surprising, as recent studies (Liu et al., 2021; Dong et al., 2021; Orvieto et al., 2023) have challenged the necessity of self-attention modules. Similar patterns can also be observed on extracted phrases with negation words, on the CNN model, and even the Transformer model in Appendix C. Conversely, the MV model demonstrates the efficacy of capturing such negation for negative adjectives, as shown in Figure 6c, demonstrating that the multiplication mechanism may play a more effective role in composing semantic meanings.

## 5.2 Discussion

Our experimental results verify our theoretical analysis of the feature extraction mechanisms employed by fundamental models during their training process. These findings are consistent even with network widths as small as $d = 64$, a scenario in which the infinite-width hypothesis is not fully realized. This observed pattern underscores the robustness and generalizability of our model, a conclusion that aligns with the insights presented by Arora et al. (2019). They suggest that as network width expands, the NTK closely approximates the computation under infinite width conditions while keeping the error within established bounds. In our study, we noted that both CNN and Self-Attention models predominantly rely on the linear combination of token-label features. However, they exhibit limi-

tations in effectively composing n-grams beyond tokens, a deficiency highlighted in negation cases. This observation points towards a potential necessity for alternative models that are adept at handling tasks involving complex n-gram features. This observation aligns with studies by Bhattamishra et al. (2020); Hahn (2020); Yao et al. (2021); Chiang and Cholak (2022), which underscore the constraints of self-attention modules, despite their practical successes. Contrarily, the MV model, based on matrix-vector multiplication, can better capture such negation evident in both analytical and observational perspectives. This model emerges as a promising alternative for tasks that hinge on the interpretation of n-grams. Regarding activation functions, our findings indicate that the utilization of *ReLU* does not significantly impact the nature of features learned but can introduce feature bias. Consequently, we suggest the exploration of alternative activation functions to mitigate this bias, enhancing the model's performance and reliability in diverse applications.

## 6 Conclusions

We propose a theoretical approach to delve into the feature extraction mechanisms behind neural models. By focusing on the learning dynamics of neural models under extreme conditions, we can shed light on useful features acquired from training data. We apply our approach to several fundamental models for text classification and explain how these models acquire features during gradient descent. Meanwhile, our approach allows us to reveal significant factors for feature extraction. For example, inappropriate choice of activation functions may induce a feature bias. Furthermore, we may also infer the limitations of a model based on the features it acquires, thereby aiding in the selection (or design) of an appropriate model for specific downstream tasks. Despite the infinite-width hypothesis, the patterns observed are remarkable with finite widths. Our future directions include analyzing more complex neural architectures.

---

[8]The negation word *not* appears more frequently in negative instances (2086 times compared to 813 in positive instances).

## Limitations

Despite the findings on the aforementioned fundamental models, applying our approach to analyze complex models like Transformers, which incorporate numerous layers, non-linear activation functions, and normalizations, presents challenges due to the increased complexity. These factors contribute to more intricate learning dynamics, making it less straightforward to gain comprehensive insights into the model's behavior. We would like to investigate and formulate them in future directions.

## Acknowledgements

We would like to thank the anonymous reviewers, our meta-reviewer, and senior area chairs for their constructive comments and support on this work. This research/project is supported by Ministry of Education, Singapore, under its Tier 3 Programme (The Award No.: MOET320200004), and Ministry of Education, Singapore, under its Academic Research Fund (AcRF) Tier 2 Programme (MOE AcRF Tier 2 Award No. : MOE-T2EP20122-0011). Any opinions, findings and conclusions or recommendations expressed in this material are those of the authors and do not reflect the views of the Ministry of Education, Singapore.

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

## A  Learning Dynamics of Models with Infinite-width

### A.1  MLP model

The representation for the instance $x$ is defined as

$$\boldsymbol{h} = \sum_{j=1}^{l^{(x)}} \frac{1}{\sqrt{d}} \phi(\boldsymbol{W} \frac{1}{\sqrt{d}} \boldsymbol{W}^e \boldsymbol{e}_j), \qquad (22)$$

and the label score of $x$ is computed as follows,

$$s = \boldsymbol{v}^\top \boldsymbol{h} = \sum_{j=1}^{l^{(x)}} \frac{\boldsymbol{v}^\top}{\sqrt{d}} \phi(\boldsymbol{W} \frac{1}{\sqrt{d}} \boldsymbol{W}^e \boldsymbol{e}_j), \quad (23)$$

where $l^{(x)}$ is the instance length, $\boldsymbol{W} \in \mathbb{R}^{d_{out} \times d_{in}}$ is the weight of the hidden layer, $\boldsymbol{W}^e \in \mathbb{R}^{d_{in} \times |V|}$ is the weight of the embedding layer, and $\boldsymbol{v} \in \mathbb{R}^{d_{out}}$ is the final layer weight. For simplicity, we let $d_{out} = d_{in} = d$. $\phi$ is the element-wise *ReLU* function. $\boldsymbol{e}_j$ is the *one-hot* vector for token $e_j$.

The gradients of the parameters can be computed as follows,

$$\begin{aligned} \frac{\partial s}{\partial \boldsymbol{v}} &= \sum_{j=1}^{l^{(x)}} \frac{1}{\sqrt{d}} \phi(\boldsymbol{c}_j), \\ \frac{\partial s}{\partial \boldsymbol{W}} &= \sum_{j=1}^{l^{(x)}} \frac{\boldsymbol{D}_j \boldsymbol{v}}{d} (\boldsymbol{W}^e \boldsymbol{e}_j)^\top, \\ \frac{\partial s}{\partial \boldsymbol{W}^e} &= \sum_{j=1}^{l^{(x)}} \frac{\boldsymbol{W}^\top \boldsymbol{D}_j \boldsymbol{v}}{d} \boldsymbol{e}_j^\top. \end{aligned} \qquad (24)$$

where

$$\begin{aligned} \boldsymbol{c}_j &= \boldsymbol{W} \frac{1}{\sqrt{d}} \boldsymbol{W}^e \boldsymbol{e}_j, \\ \boldsymbol{D}_j &= \frac{\partial \phi(\boldsymbol{c}_j)}{\partial \boldsymbol{c}_j}. \end{aligned} \qquad (25)$$

Note that $\boldsymbol{D}_j$ is a diagonal matrix with elements being either 1s or 0s.

Given a test input $x'$, the learning dynamics of the label score $s'$ will be

$$\dot{s}' = \frac{1}{m} \sum_{(x,y) \in \mathcal{D}} g(-ys^{(x)}) y \Theta(x', x). \qquad (26)$$

With the gradients, we can obtain the NTK

$\Theta(x', x)$ for this MLP model as follows,

$$
\begin{aligned}
\Theta(x', x) = &\frac{1}{d} \sum_{i=1}^{l^{(x')}} \sum_{j=1}^{l^{(x)}} \boldsymbol{\phi}^\top(\boldsymbol{c}_i) \boldsymbol{\phi}(\boldsymbol{c}_j) \\
&+ \frac{1}{d^2} \sum_{i=1}^{l^{(x')}} \sum_{j=1}^{l^{(x)}} \boldsymbol{v}^\top \boldsymbol{D}_i \boldsymbol{D}_j \boldsymbol{v} \boldsymbol{e}_i^\top (\boldsymbol{W}^e)^\top \boldsymbol{W}^e \boldsymbol{e}_j \\
&+ \frac{1}{d^2} \sum_{i=1}^{l^{(x')}} \sum_{j=1}^{l^{(x)}} \boldsymbol{v}^\top \boldsymbol{D}_i \boldsymbol{W} (\boldsymbol{W})^\top \boldsymbol{D}_j \boldsymbol{v} \boldsymbol{e}_i^\top \boldsymbol{e}_j,
\end{aligned}
\tag{27}
$$

where

$$
\begin{aligned}
\boldsymbol{c}_i &= \boldsymbol{W} \frac{1}{\sqrt{d}} \boldsymbol{W}^e \boldsymbol{e}_i, \\
\boldsymbol{D}_i &= \frac{\partial \phi(\boldsymbol{c}_i)}{\partial \boldsymbol{c}_i}.
\end{aligned}
\tag{28}
$$

The label score of an instance can be viewed as the sum of the label scores of all the tokens and the NTK can be viewed as the sum of the interaction between each token pair from the test input $x'$ and the training instance $x$.

### A.1.1 NTKs under the Infinite-width

It's delicate to analyze such an NTK directly in practice as the NTK will vary over time. However, the previous work discussed in the literature has proved that the NTK will converge and stay constant during training under the infinite-width condition. Now let us consider the infinite-width scenario (Lee et al., 2018) and obtain the NTK subsequently. We first give the NTK between two instances and then give the NTK between an input token and an instance.

**NTK between instances** Assume we initialize parameters following Gaussian distributions, i.e., $\boldsymbol{W}_{ij} \sim \mathcal{N}(0, \sigma_w^2)$, $\boldsymbol{W}_{ij}^e \sim \mathcal{N}(0, \sigma_e^2)$, and $\boldsymbol{v}_j \sim \mathcal{N}(0, \sigma_v^2)$.

**Lemma A.1.** When the network width approaches infinity, $\Theta(x', x)$ converges to a deterministic NTK $\Theta_\infty(x', x)$ during training which obeys

$$
\begin{aligned}
&\Theta_\infty(x', x) \\
&= \sum_{i=1}^{l^{(x')}} \sum_{j=1}^{l^{(x)}} \frac{\sigma_e^2 \sigma_w^2}{2\pi} [\sin \alpha_{ij} + (\pi - \alpha_{ij}) \cos \alpha_{ij}] \\
&+ \sum_{i=1}^{l^{(x')}} \sum_{j=1}^{l^{(x)}} (\sigma_e^2 \sigma_v^2 + \sigma_w^2 \sigma_v^2) \frac{\boldsymbol{e}_i^\top \boldsymbol{e}_j}{2}
\end{aligned}
$$

where $\alpha_{ij} = \cos^{-1} \boldsymbol{e}_i^\top \boldsymbol{e}_j$.

We will give the proof along with the proof of Lemma 4.1, where the test input is simply a token $e$.

*Proof.* We only need to compute the NTK when the network width grows to infinity, as the NTK's convergence during training has been proved in the work of Jacot et al. (2018); Yang and Littwin (2021).

For the first part in Equation 27, the dot-product between two activation outputs can be written as,

$$
\boldsymbol{\phi}^\top(\boldsymbol{c}_i) \boldsymbol{\phi}(\boldsymbol{c}_j) = \sum_{r=1}^{d} \phi(\boldsymbol{c}_{ir}) \phi(\boldsymbol{c}_{jr}),
\tag{29}
$$

where $r$ refers to the row index and

$$
\begin{aligned}
\boldsymbol{c}_{ir} &= \boldsymbol{W}_r \frac{1}{\sqrt{d}} \boldsymbol{W}^e \boldsymbol{e}_i \\
\boldsymbol{c}_{jr} &= \boldsymbol{W}_r \frac{1}{\sqrt{d}} \boldsymbol{W}^e \boldsymbol{e}_j.
\end{aligned}
\tag{30}
$$

As elements in $\boldsymbol{W}$ and $\boldsymbol{W}^e$ follow Gaussian distributions, when the network width $d \to \infty$, $\boldsymbol{c}_{ir}$ and $\boldsymbol{c}_{jr}$ will also be Gaussian distributed respectively and they follow a Gaussian process (Lee et al., 2018). Based on the work of Cho and Saul (2009), the covariance $\mathcal{K}(\phi(\boldsymbol{c}_{ir}), \phi(\boldsymbol{c}_{jr}))$ (regardless of $r$) will be calculated as

$$
\begin{aligned}
&\mathcal{K}(\phi(\boldsymbol{c}_{ir}), \phi(\boldsymbol{c}_{jr})) \\
&= \frac{\sigma_e^2 \sigma_w^2}{2\pi} [\sin \alpha_{ij} + (\pi - \alpha_{ij}) \cos \alpha_{ij}],
\end{aligned}
\tag{31}
$$

where $\alpha_{ij} = \cos^{-1} \boldsymbol{e}_i^\top \boldsymbol{e}_j$.

Then, we arrive at

$$
\begin{aligned}
&\lim_{d \to \infty} \frac{1}{d} \boldsymbol{\phi}^\top(\boldsymbol{c}_i) \boldsymbol{\phi}(\boldsymbol{c}_j) = \mathbb{E}[\phi(\boldsymbol{c}_{ir}) \phi(\boldsymbol{c}_{jr}] \\
&= \frac{\sigma_e^2 \sigma_w^2}{2\pi} [\sin \alpha_{ij} + (\pi - \alpha_{ij}) \cos \alpha_{ij}].
\end{aligned}
\tag{32}
$$

For the second part in Equation 27, let us look at $\boldsymbol{e}_i^\top (\boldsymbol{W}^e)^\top \boldsymbol{W}^e \boldsymbol{e}_j$ first. Let $\boldsymbol{M}^e = (\boldsymbol{W}^e)^\top \boldsymbol{W}^e$ and the elements of $\boldsymbol{M}^e$ can be computed as

$$
\boldsymbol{M}_{ij}^e = \sum_{r=1}^{d} \boldsymbol{W}_{ri}^e \boldsymbol{W}_{rj}^e,
\tag{33}
$$

where $\boldsymbol{W}_{ri}^e, \boldsymbol{W}_{rj}^e \sim \mathcal{N}(0, \sigma_e^2)$. When $d \to \infty$,

$$
\begin{aligned}
\lim_{d \to \infty} \frac{1}{d} \boldsymbol{M}_{ij}^e &= \lim_{d \to \infty} \frac{1}{d} \sum_{r=1}^{d} \boldsymbol{W}_{ri}^e \boldsymbol{W}_{rj}^e = \mathbb{E}[\boldsymbol{W}_{ri}^e \boldsymbol{W}_{rj}^e] \\
&= \begin{cases} \sigma_e^2, & \text{if } i \equiv j \\ 0, & \text{otherwise} \end{cases}
\end{aligned}
\tag{34}
$$

We can thereby arrive at

$$\lim_{d\to\infty} \frac{1}{d} \boldsymbol{e}_i^\top (\boldsymbol{W}^e)^\top \boldsymbol{W}^e \boldsymbol{e}_j = \sigma_e^2 \boldsymbol{e}_i^\top \boldsymbol{e}_j. \qquad (35)$$

Let us look at $\boldsymbol{v}^\top \boldsymbol{D}_i \boldsymbol{D}_j \boldsymbol{v}$. If $\boldsymbol{e}_i^\top \boldsymbol{e}_j = 0$, which means $\boldsymbol{D}_i \neq \boldsymbol{D}_j$, we will arrive at

$$\lim_{d\to\infty} \frac{1}{d^2} \boldsymbol{v}^\top \boldsymbol{D}_i \boldsymbol{D}_j^\top \boldsymbol{v} \frac{1}{d} \boldsymbol{e}_i^\top (\boldsymbol{W}^e)^\top \boldsymbol{W}^e \boldsymbol{e}_j$$
$$= \lim_{d\to\infty} \frac{1}{d} \sum_{r=1}^d \boldsymbol{D}_{irr} \boldsymbol{D}_{jrr} \boldsymbol{v}_r^2 \sigma_e^2 \boldsymbol{e}_i^\top \boldsymbol{e}_j \qquad (36)$$
$$= 0,$$

where $\boldsymbol{D}_{irr}$ and $\boldsymbol{D}_{jrr}$ are diagonal elements in $\boldsymbol{D}_i$ and $\boldsymbol{D}_j$ respectively. $\boldsymbol{v}_r^2$ is the $r$-th element in $\boldsymbol{v}$.

If $\boldsymbol{e}_i^\top \boldsymbol{e}_j \neq 0$, which means $\boldsymbol{D}_i = \boldsymbol{D}_j$, we will have

$$\lim_{d\to\infty} \frac{1}{d} \boldsymbol{v}^\top \boldsymbol{D}_i \boldsymbol{D}_j \boldsymbol{v} \frac{1}{d} \boldsymbol{e}_i^\top (\boldsymbol{W}^e)^\top \boldsymbol{W}^e \boldsymbol{e}_j$$
$$= \frac{1}{2} \sigma_v^2 \sigma_e^2 \boldsymbol{e}_i^\top \boldsymbol{e}_j. \qquad (37)$$

Similarly, we can get the third part in Equation 27,

$$\lim_{d\to\infty} \frac{1}{d^2} \sum_{i=1}^{l^{(x')}} \sum_{j=1}^{l^{(x)}} \boldsymbol{v}^\top \boldsymbol{D}_i \boldsymbol{W}(\boldsymbol{W})^\top \boldsymbol{D}_j \boldsymbol{v} \boldsymbol{e}_i^\top \boldsymbol{e}_j$$
$$= \frac{1}{2} \sigma_v^2 \sigma_w^2 \boldsymbol{e}_i^\top \boldsymbol{e}_j. \qquad (38)$$

Plugging the above equations in Equation 27, we will arrive at Lemma A.1. $\qquad\square$

**NTK between a token and an instance** Next, we give the proof of Lemma 4.1 based on Lemma A.1.

*Proof.* Note that as $\boldsymbol{e}_i$ and $\boldsymbol{e}_j$ are one-hot vectors, their dot-product $\boldsymbol{e}_i^\top \boldsymbol{e}_j$ satisfies that, $\boldsymbol{e}_i^\top \boldsymbol{e}_j = 1$ when $\boldsymbol{e}_i \equiv \boldsymbol{e}_j$ or 0 otherwise. Therefore, $\alpha_{ij} = \cos^{-1} \boldsymbol{e}_i^\top \boldsymbol{e}_j$, $\alpha_{ij}$ can be $\frac{\pi}{2}$ or 0. And the kernel can be further written as

$$\Theta_\infty(x', x) = \sum_{i=1}^{l^{(x')}} \sum_{j=1}^{l^{(x)}} \rho \boldsymbol{e}_i^\top \boldsymbol{e}_j + \sum_{i=1}^{l^{(x')}} \sum_{j=1}^{l^{(x)}} \mu \qquad (39)$$

where $\rho = \frac{(\pi-1)\sigma_e^2 \sigma_w^2}{2\pi} + \frac{\sigma_e^2 \sigma_v^2 + \sigma_w^2 \sigma_v^2}{2}$ and $\mu = \frac{\sigma_e^2 \sigma_w^2}{2\pi}$. This means the converged kernel $\Theta_\infty(x', x)$ will keep being *non-negative* during training, and the direction of the dynamics will depend on the label $y$ in Equation 26.

Let us look at the *token-label features* learned in the dynamics. As previously mentioned, the instance label score can be viewed as the sum of token label scores. Consider the scenario where the test input $x'$ is simply a token $e$, the NTK $\Theta_\infty(e, x)$ obeys

$$\Theta_\infty(e, x) = \sum_{j=1}^{l^{(x)}} \rho \boldsymbol{e}^\top \boldsymbol{e}_j + \sum_{j=1}^{l^{(x)}} \mu, \qquad (40)$$

where the dot-product $\sum_{j=1}^{l^{(x')}} \boldsymbol{e}^\top \boldsymbol{e}_j$ will be interpreted as the frequency of $e$ appearing in instance $x$. We can thereby arrive at Lemma 4.1. $\qquad\square$

### A.1.2 Features Encoded in Gradient Descent

With Lemma 4.1 and Equation 5, we can get Theorem 4.2. Under the infinite-width, the dynamics of token $e$'s label score obey

$$\dot{s}^e = \underbrace{\frac{1}{m} \sum_{(x,y)\in\mathcal{D}} g(-ys^{(x)}) y \rho \omega(e, x)}_{A}$$
$$+ \underbrace{\frac{1}{m} \sum_{(x,y)\in\mathcal{D}} g(-ys^{(x)}) y \mu l^{(x)}}_{B}, \qquad (41)$$

where $\omega(e, x)$ is the frequency of token $e$ in instance $x$. $\omega(e, x)$ depends on the training data and will not change over time. We cannot give a *closed-form* solution for this ODE due to the *non-linearity* of the sigmoid function $g(-ys)$. However, as $g(-ys^{(x)})$ is *non-negative*, there can be certain interesting trends for the label scores.

Note that the first term $A$ in Equation 41 will depend on this token's *term-frequencies* $\omega(e, x)$ in each training instance. The second term $B$ depends on the entire training set and is shared by all the tokens $e$. For example, if $e$ does not appear in an instance $x$, $\omega(e, x)$ will be 0.

### A.1.3 Bias Induced in Gradient Descent

Let us look at the term $B$ in Equation 41, which can be viewed as an induced *feature bias* shared by all tokens. It is affected by the variances and the instance lengths. Suppose the term $B$ is sufficiently large; in this case, the positive tokens and the negative tokens will be affected by this bias in different directions during training. For example, if term $B$ is positively large, it will positively contribute to the learning dynamics of positive tokens and make their label scores much larger than 0 after sufficient updates. However, the negative tokens will have

weakened learning dynamics and end up with label scores close to 0.

In Equation 41, both $\omega(e, x)$ and $l^{(x)}$ are determined by the training instances. Therefore, the factor $\rho$ and $\mu$ defined in Equation 39 can affect the influence of this induced bias. It can be inferred that, a significantly large variance $\sigma_v$ can make $\rho$ much larger than $\mu$, thus reducing the influence of the bias.

## A.2 CNN Model

We consider the 1-dimension CNN, which is commonly used in NLP tasks. The kernel size, stride size, and padding size will be set as $K$, 1, and $K - 1$, respectively.

For each sliding window $c_j$ that consists of $K$ consecutive tokens, the corresponding feature $c_j \in \mathbb{R}^{d_{out}}$ can be represented as

$$c_j = \sum_{k=1}^{K} W_k^c \frac{1}{\sqrt{d}} W^e e_{j+k-1}, \qquad (42)$$

where $W_k^c \in \mathbb{R}^{d_{out} \times d_{in}}$ is the kernel weight corresponding to the $k$-th token in the sliding window, $W^e \in \mathbb{R}^{d_{in} \times V}$ is the embedding matrix, and $e$ is the one-hot vector for token $e$. We also let $d_{in} = d_{out} = d$.

Given an input $x$, the label score will be calculated as

$$s = \frac{v^\top}{\sqrt{d}} \sum_{j=-(K-1)}^{l^{(x)}} \phi(c_j), \qquad (43)$$

where $-(K-1)$ means the position for the leftmost padding token. The first $K-1$ and last $K-1$ tokens in an instance are padding ones represented by zero vectors. $\phi$ is the element-wise *ReLU* function.

For brevity, we will denote $\sum_{j=-(K-1)}^{l^{(x)}}$ by $\sum_j$. The gradients will be computed as

$$\frac{\partial s}{\partial v} = \sum_j \frac{1}{\sqrt{d}} \phi\left(\sum_{k=1}^{K} W_k^c \frac{1}{\sqrt{d}} W^e e_{j+k-1}\right),$$
$$\frac{\partial s}{\partial W_k^c} = \sum_j \frac{D_j v}{d} (W^e e_{j+k-1})^\top,$$
$$\frac{\partial s}{\partial W^e} = \sum_j \sum_{k=1}^{K} \frac{(W_k^c)^\top D_j v}{d} e_{j+k-1}^\top,$$

$$(44)$$

where

$$c_j = \sum_{k=1}^{K} W_k^c \frac{1}{\sqrt{d}} W^e e_{j+k-1}, \qquad (45)$$

$$D_j = \frac{\partial \phi(c_j)}{\partial c_j}. \qquad (46)$$

For a test input $x'$, the dynamics of its label score will obey

$$\dot{s}_t' = \frac{1}{m} \sum_{(x,y) \in \mathcal{D}} y g(-y s_t) \Theta(x', x), \qquad (47)$$

where

$$\Theta(x', x) = \sum_i \sum_j \underbrace{\frac{1}{d} \phi^\top(c_i) \phi(c_j)}_{A} +$$
$$\sum_{i,j,k} \underbrace{\frac{v^\top}{d^2} D_i D_j v e_{i+k-1}^\top (W^e)^\top W^e e_{j+k-1}}_{B} +$$
$$\sum_{i,j,k,k'} \underbrace{\frac{v^\top}{d^2} D_i W_{k'}^c (W_k^c)^\top D_j v e_{i+k-1}^\top e_{j+k-1}}_{C},$$

$$(48)$$

where $\sum_{i,j,k} = \sum_i \sum_j \sum_{k=1}^{K}$ and $\sum_{i,j,k,k'} = \sum_i \sum_j \sum_{k'=1}^{K} \sum_{k=1}^{K}$

### A.2.1 NTK under the Infinite-width

It should be highlighted that, even when the network width approaches infinity, it may not be easy to describe the converged NTK with an explicit closed-form expression due to the integrals used in obtaining expectations. However, we will show that the converged NTK can be written as functions of parameter variances and similarity between sliding windows, and the functions are affected by the similarity between sliding windows.

**Lemma A.2.** Assume we initialize parameters following Gaussian distributions, i.e., $W_{ij}^c \sim \mathcal{N}(0, \sigma_w^2)$, $W_{ij}^e \sim \mathcal{N}(0, \sigma_e^2)$, and $v_j \sim \mathcal{N}(0, \sigma_v^2)$ ($W_{ij}^c$ refers to an element in $W_k^c$). When the network width approaches infinity, given two instances $x'$ and $x$, the NTK $\Theta(x', x)$ for the CNN model converges to

$$\Theta_\infty(x', x) = \sum_i \sum_j F(\omega_c(c_i, c_j))$$
$$+ \sum_i \sum_j \sigma_v^2 \sigma_e^2 H(\omega_c(c_i, c_j)) \omega_2(c_i, c_j)$$
$$+ \sum_i \sum_j \sigma_v^2 \sigma_w^2 H(\omega_c(c_i, c_j)) \omega_2(c_i, c_j),$$

where $c_i$ and $c_j$ are sliding windows starting from the $i$-th token and the $j$-th token in instances $x'$ and $x$, respectively. Functions $\omega_c$ and $\omega_2$ are defined as

$$\omega_c(c_i, c_j) = \sum_{k'=1}^{K} \boldsymbol{e}_{i+k'-1}^{\top} \sum_{k=1}^{K} \boldsymbol{e}_{j+k-1}$$

$$\omega_2(c_i, c_j) = \sum_{k=1}^{K} \boldsymbol{e}_{i+k-1}^{\top} \boldsymbol{e}_{j+k-1}.$$

Functions $F$ and $H$ are defined as

$$F(n) =$$
$$\mathbb{E}[\phi(\sum_{i=1}^{n} w_i + \sum_{i=n+1}^{K} w_i)\phi(\sum_{i=1}^{n} w_i + \sum_{i=n+1}^{K} w_i')],$$

$$H(n) =$$
$$\mathbb{E}[D(\sum_{i=1}^{n} w_i + \sum_{i=n+1}^{K} w_i)D(\sum_{i=1}^{n} w_i + \sum_{i=n+1}^{K} w_i')],$$

where $n$ ($0 \leq n \leq K$) is the number of tokens shared by two sliding windows and $w_i, w_i' \sim \mathcal{N}(0, \sigma_e^2 \sigma_w^2)$. $\phi$ and $D$ are the *ReLU* function and *step* function[9] respectively.

*Remark.* It can be seen that the similarity between sliding windows influences the converged NTK. As the variances are constants, we can focus on $\omega_c$ and $\omega_2$, which can be viewed as similarity metrics for sliding windows. The former does not take positional information into consideration, while the latter does. Particularly, we can have that $\omega_c(c_i, c_j) \geq \omega_2(c_i, c_j)$. When the two sliding windows share tokens in the right order, $\omega_2(c_i, c_j)$ becomes large.

**Proof of Lemma A.2** We give the proofs for each part in the NTK shown in Equation 48. Let us prove that the *ReLU* output multiplication (term $A$ in Equation 48) can be written as a function of $F(n)$ under the infinite-width condition.

*Proof.* The *ReLU* output multiplication can be written as

$$\frac{1}{d}\boldsymbol{\phi}^{\top}(\boldsymbol{c}_i)\boldsymbol{\phi}(\boldsymbol{c}_j) = \frac{1}{d}\sum_{r=1}^{d} \phi(\boldsymbol{c}_{ir})\phi(\boldsymbol{c}_{jr}), \quad (49)$$

where $r$ refers to the $r$-th element and

$$\boldsymbol{c}_{ir} = \sum_{k=1}^{K} \boldsymbol{W}_{kr}^{c} \frac{1}{\sqrt{d}} \boldsymbol{W}^{e} \boldsymbol{e}_{i+k-1},$$
$$\boldsymbol{c}_{jr} = \sum_{k=1}^{K} \boldsymbol{W}_{kr}^{c} \frac{1}{\sqrt{d}} \boldsymbol{W}^{e} \boldsymbol{e}_{j+k-1}. \quad (50)$$

[9] $D(x) = 1$ if $x > 0$, 0 otherwise.

$\boldsymbol{W}_{kr}^{c}$ is the $r$-th row of matrix $\boldsymbol{W}_{k}^{c}$. Elements in $\boldsymbol{W}_{k}^{c}$ and $\boldsymbol{W}^{e}$ follow Gaussian distributions and are I.I.D (Independent and identically distributed) random variables. With the infinite network width, given a token $e$, elements in $\boldsymbol{W}_{k}^{c} \frac{1}{\sqrt{d}} \boldsymbol{W}^{e} e$ can also be viewed as I.I.D and follow a Gaussian process (Lee et al., 2018). Let $w = \boldsymbol{W}_{kr}^{c} \frac{1}{\sqrt{d}} \boldsymbol{W}^{e} e$. We can get that $w \sim \mathcal{N}(0, \sigma_e^2 \sigma_w^2)$. Similarly, we can obtain that $\boldsymbol{c}_{ir}$ and $\boldsymbol{c}_{jr}$ follow Gaussian distributions.

When the network width approaches infinity, the multiplication will be viewed as the expectation as follows,

$$\lim_{d\to\infty} \frac{1}{d}\boldsymbol{\phi}^{\top}(\boldsymbol{c}_i)\boldsymbol{\phi}(\boldsymbol{c}_j) = \mathbb{E}[\phi(\boldsymbol{c}_{ir})\phi(\boldsymbol{c}_{jr})]. \quad (51)$$

Then, we can arrive at

$$\lim_{d\to\infty} \frac{1}{d}\boldsymbol{\phi}^{\top}(\boldsymbol{c}_i)\boldsymbol{\phi}(\boldsymbol{c}_j) =$$
$$\mathbb{E}[\phi(\sum_{i=1}^{n} w_i + \sum_{i=n+1}^{K} w_i)\phi(\sum_{i=1}^{n} w_i + \sum_{i=n+1}^{K} w_i')] = F(n),$$
$$(52)$$

where $n$ is the number of shared tokens between sliding windows $c_i$ and $c_j$.

$\square$

We prove term $B$ in Equation 48 can be written as a function of $H(n)$ under the infinite-width condition.

*Proof.* With the infinite network width, we can have

$$\lim_{d\to\infty} \frac{1}{d}\boldsymbol{v}^{\top}\boldsymbol{D}_i\boldsymbol{D}_j^{\top}\boldsymbol{v} = \lim_{d\to\infty} \frac{1}{d}\sum_{r=1}^{d} v_r^2 \boldsymbol{D}_{irr}\boldsymbol{D}_{jrr}$$
$$= \sigma_v^2 \mathbb{E}[\boldsymbol{D}_{irr}\boldsymbol{D}_{jrr}]$$
$$\lim_{d\to\infty} \frac{1}{d}\boldsymbol{e}_{i+k-1}^{\top}(\boldsymbol{W}^e)^{\top}\boldsymbol{W}^e \boldsymbol{e}_{j+k-1}$$
$$= \sigma_e^2 \boldsymbol{e}_{i+k-1}^{\top}\boldsymbol{e}_{j+k-1},$$
$$(53)$$

where $r$ is the row number. $\boldsymbol{D}_{irr}$ and $\boldsymbol{D}_{jrr}$ refer to the $r$-th elements in the diagonal positions of $\boldsymbol{D}_i$ and $\boldsymbol{D}_j$, respectively.

Term $B$ will obey

$$\lim_{d\to\infty} \sum_{k=1}^{K} \frac{\boldsymbol{v}^{\top}}{d^2}\boldsymbol{D}_i\boldsymbol{D}_j^{\top}\boldsymbol{v}\boldsymbol{e}_{i+k-1}^{\top}(\boldsymbol{W}^e)^{\top}\boldsymbol{W}^e \boldsymbol{e}_{j+k-1}$$
$$= \sigma_v^2 \sigma_e^2 \mathbb{E}[\boldsymbol{D}_{irr}\boldsymbol{D}_{jrr}]\boldsymbol{e}_{i+k-1}^{\top}\boldsymbol{e}_{j+k-1}.$$
$$(54)$$

We can compute the expectation $\mathbb{E}[\boldsymbol{D}_{irr}\boldsymbol{D}_{jrr}]$ (outputs of the step function) similarly to that of *ReLU* output multiplication.

Let $w_i \sim \mathcal{N}(0, \sigma_e^2 \sigma_w^2)$. $\boldsymbol{D}_{irr}$ ($\boldsymbol{D}_{jrr}$) equals 1 when the corresponding $\boldsymbol{c}_{ir} > 0$ ($\boldsymbol{c}_{jr} > 0$), 0 otherwise. Then, we can obtain

$$
\begin{aligned}
&\mathbb{E}[\boldsymbol{D}_{irr}\boldsymbol{D}_{jrr}] \\
&= \mathbb{E}[D(\sum_{i=1}^{n} w_i + \sum_{i=n+1}^{K} w_i)D(\sum_{i=1}^{n} w_i + \sum_{i=n+1}^{K} w_i')] \\
&= H(n),
\end{aligned}
\tag{55}
$$

where $n$ is the number of shared tokens between sliding windows $c_i$ and $c_j$.

For the third part (term $C$ in Equation 48), when the network width approaches infinity, the expectation will be

$$
\begin{aligned}
&\lim_{d \to \infty} \frac{\boldsymbol{v}^\top}{d^2} \boldsymbol{D}_i \boldsymbol{W}_{k'}^c (\boldsymbol{W}_k^c)^\top \boldsymbol{D}_j^\top \boldsymbol{v} \boldsymbol{e}_i^\top \boldsymbol{e}_j \\
&= \sigma_v^2 \sigma_w^2 \mathbb{E}[\boldsymbol{D}_{irr}\boldsymbol{D}_{jrr}]\boldsymbol{e}_i^\top \boldsymbol{e}_j.
\end{aligned}
\tag{56}
$$

It is obvious that this term will be positive if the two windows $c_i$ and $c_j$ do not share any tokens, 0 otherwise. $\square$

Now, let us look at the $F$ and $H$ functions, which have interesting properties regarding the similarities between sliding windows.

**Proposition A.1.** *Both $F$ and $H$ functions in Lemma A.2 are monotonically increasing as the on-negative integer $n$ increases. Given two non-negative integers $n$ and $n'$, when $n' > n$, the two functions obey*

$$
\begin{aligned}
F(n') &\geq F(n), \\
H(n') &\geq H(n).
\end{aligned}
$$

*Remark.* This indicates the more similar the two sliding windows are (i.e., the more tokens they share), the larger $F$ and $H$ will be.

The core idea leveraged in the proofs is based on the inequality, $Var(x) = \mathbb{E}(x^2) - \mathbb{E}(x)^2 \geq 0$, where $x$ is a random variable yielding to a Gaussian distribution.

**Monotonicity of $F$ function**

*Proof.* First, let us consider the scenario $n > 0$, which means the two sliding windows share tokens.

The expectation can be computed as

$$
\begin{aligned}
&F(n) = \\
&\mathbb{E}[\phi(\sum_{i=1}^{n} w_i + \sum_{i=n+1}^{K} w_i)\phi(\sum_{i=1}^{n} w_i + \sum_{i=n+1}^{K} w_i')] \\
&= \int_{-\infty}^{\infty} \phi(z_{1:n} + z_{n+1:K})\phi(z_{1:n} + z_{n+1:K}') \\
&\quad p(w_1) \ldots p(w_K)p(w_{n+1}') \ldots p(w_K') \\
&\quad dw_1 \ldots dw_K dw_{n+1}' \ldots dw_K' \\
&= \mathbb{E}[\psi(w_1 \ldots w_n)^2],
\end{aligned}
\tag{57}
$$

where

$$
\begin{aligned}
\psi(w_1 \ldots w_n) &= \int_{-\infty}^{\infty} \phi(z_{1:n} + z_{n+1:K})p(w_{n+1}) \\
&\quad \ldots p(w_K)dw_{n+1} \ldots dw_K \\
z_{1:n} &= \sum_{i=1}^{n} w_i \\
z_{n+1:K} &= \sum_{i=n+1}^{K} w_i \\
z_{n+1:K}' &= \sum_{i=n+1}^{K} w_i',
\end{aligned}
\tag{58}
$$

and $\int_{-\infty}^{\infty}$ refers to the integrals for all the variables involved.

The expectation for the multiplication of two sliding windows without sharing tokens can be written as

$$
\begin{aligned}
&F(0) = \\
&\mathbb{E}[\phi(\sum_{i=1}^{n} w_i + \sum_{i=n+1}^{K} w_i)\phi(\sum_{i=1}^{n} w_i' + \sum_{i=n+1}^{K} w_i')] \\
&= \mathbb{E}[\psi(w_1 \ldots w_n)]^2,
\end{aligned}
\tag{59}
$$

which can be described as

$$
\begin{aligned}
&F(n) = \\
&\mathbb{E}[\phi(\sum_{i=1}^{n} w_i + \sum_{i=n+1}^{K} w_i)\phi(\sum_{i=1}^{n} w_i + \sum_{i=n+1}^{K} w_i')] \\
&\geq F(0) = \\
&\mathbb{E}[\phi(\sum_{i=1}^{n} w_i + \sum_{i=n+1}^{K} w_i)\phi(\sum_{i=1}^{n} w_i' + \sum_{i=n+1}^{K} w_i')],
\end{aligned}
\tag{60}
$$

where $w_i$ and $w_i'$ are I.I.D random variables and $n \geq 1$. This indicates that when two sliding windows share tokens, the expectation will be larger than that in the case where two sliding windows do not share any tokens.

We can prove that one more shared tokens between two sliding windows can result in an increase in expectation. Let us increase the number of shared tokens by 1 between the two sliding windows, the expectation can be written as

$$F(n+1) =$$
$$\mathbb{E}[\phi(\sum_{i=1}^{n+1} w_i + \sum_{i=n+2}^{K} w_i)\phi(\sum_{i=1}^{n+1} w_i + \sum_{i=n+2}^{K} w_i')]$$
$$= \int_{-\infty}^{\infty} \phi(z_{1:n+1} + z_{n+2:K})\phi(z_{1:n+1} + z_{n+2:K}')$$
$$p(w_1)\ldots p(w_K)p(w_{n+2}')\ldots p(w_K')$$
$$dw_1 \ldots dw_K dw_{n+2}' \ldots dw_K'$$
$$=$$
$$\int_{-\infty}^{\infty} \left[ \int_{-\infty}^{\infty} \xi(w_1,\ldots,w_{n+1})^2 p(w_{n+1}) dw_{n+1} \right]$$
$$p(w_1)\ldots p(w_n) dw_1 \ldots dw_n,$$
(61)

where $\xi(w_1,\ldots,w_{n+1}) = \int_{-\infty}^{\infty} \phi(z_{1:n+1} + z_{n+2:K})p(w_{n+2})\ldots p(w_K)dw_{n+2}\ldots dw_K$.

Note that the expectation $\mathbb{E}[\phi(\sum_{i=1}^{n} w_i + \sum_{i=n+1}^{K} w_i)\phi(\sum_{i=1}^{n} w_i + \sum_{i=n+1}^{K} w_i')]$ can be written as

$$F(n) =$$
$$\mathbb{E}[\phi(\sum_{i=1}^{n} w_i + \sum_{i=n+1}^{K} w_i)\phi(\sum_{i=1}^{n} w_i + \sum_{i=n+1}^{K} w_i')]$$
$$= \int_{-\infty}^{\infty} \left[ \int_{-\infty}^{\infty} \xi(w_1,\ldots,w_{n+1})p(w_{n+1})dw_{n+1} \right]^2$$
$$p(w_1)\ldots p(w_n)dw_1 \ldots dw_n.$$
(62)

As we have

$$\int_{-\infty}^{\infty} \xi(w_1,\ldots,w_{n+1})^2 p(w_{n+1}) dw_{n+1} \geq$$
$$\left[ \int_{-\infty}^{\infty} \xi(w_1,\ldots,w_{n+1})p(w_{n+1})dw_{n+1} \right]^2,$$
(63)

we can arrive at

$$F(n+1) =$$
$$\mathbb{E}[\phi(\sum_{i=1}^{n+1} w_i + \sum_{i=n+2}^{K} w_i)\phi(\sum_{i=1}^{n+1} w_i + \sum_{i=n+2}^{K} w_i')]$$
$$\geq F(n) =$$
$$\mathbb{E}[\phi(\sum_{i=1}^{n} w_i + \sum_{i=n+1}^{K} w_i)\phi(\sum_{i=1}^{n} w_i + \sum_{i=n+1}^{K} w_i')].$$
(64)

We can prove recursively for the case $F(n+l) \geq F(n)$ where $l > 1$. Therefore, the expectation will be monotonically increasing with $n$. $\square$

**Monotonicity of $H$ function**   We can obtain the expectation of the other terms in the NTK similarly as the activation function $\phi$ can be replaced with different activation functions with non-negative outputs.

**Proofs of Lemma 4.3 and Theorem 4.4**   With Lemma A.2 and Proposition A.1, we can prove Lemma 4.3 by replacing the test input $x'$ with sliding window $c$. Similarly, we can prove Theorem 4.4 with Lemma 4.3.

Let us focus on a single sliding window $c$. The converged NTK between a sliding window $c$ (consisting of tokens $e_1, e_2, \ldots, e_K$) and instance $x$ obeys

$$\Theta_\infty(c,x) = \sum_j F(\omega_c(c,c_j))$$
$$+ \sum_j \sigma_v^2 \sigma_e^2 H(\omega_c(c,c_j))\omega_2(c,c_j)$$
$$+ \sum_j \sigma_v^2 \sigma_w^2 H(\omega_c(c,c_j))\omega_2(c,c_j).$$
(65)

If $c$ does not share tokens with any of the sliding windows in $x$, the NTK $\Theta_\infty(c,x)$ will reach its minimum, namely, $\Theta_\infty(c,x) = \sum_j F(\omega_c(0))$. Otherwise, $\Theta_\infty(c,x)$ will be significantly large if $c$ bears similarity to the sliding windows of $x$. Then the dynamics of the label score of $c$ obey

$$\dot{s}^c = \frac{\rho}{m} \sum_{(x,y)\in\mathcal{D}} yg(-ys^{(x)})\Theta_\infty(c,x).$$

The sliding windows can be interpreted as n-grams. Suppose an n-gram represented by a sliding window $c$ bears similarity only to the sliding windows in positive (negative) instances. In that case, it will receive positively (negatively) large gains in one direction during training and will likely end up being significant.

Let us examine what features will be learned for a single token $e$. We define a positional-relevance label score as follows,

$$s_{[k]}^e = \frac{1}{\sqrt{d}} \boldsymbol{v}^\top \boldsymbol{\phi}(\frac{1}{\sqrt{d}} \boldsymbol{W}_k^c \boldsymbol{W}^e \boldsymbol{e}),$$
(66)

which reflects the label score of token $e$ in the $k$-th position of a sliding window.

It should be noted that based on our analysis, the kernel size $K$ in Equation 10 does not affect the monotonicity of the $F$ and $H$ functions. Suppose the sliding window $c$ shares tokens with instance $x$; the NTK will learn the features regardless of the kernel size $K$.

## A.3 SA Model

Let us define an intermediate score $s_j$, corresponding to the label score in the $j$-th output, as follows,

$$s_j = \boldsymbol{v}^\top \sum_{j=1}^{l^{(x)}} \frac{\alpha_{ij}}{\sqrt{d}} \boldsymbol{v}^\top \boldsymbol{W}^e \boldsymbol{e}_j. \qquad (67)$$

The gradients can be computed as

$$\frac{\partial s_i}{\partial \boldsymbol{v}} = \sum_{j=1}^{l^{(x)}} \alpha_{ij} \frac{1}{\sqrt{d}} \boldsymbol{W}^e \boldsymbol{e}_j,$$

$$\frac{\partial s_i}{\partial \boldsymbol{W}^e} = \sum_{j=1}^{l^{(x)}} \sum_{k=1}^{l^{(x)}} \alpha_{ij}(\delta_{jk} - \alpha_{ik}) \frac{\boldsymbol{v}^\top \boldsymbol{W}^e \boldsymbol{e}_j}{d\sqrt{d}}$$
$$[\boldsymbol{W}^e \boldsymbol{e}_k \boldsymbol{e}_i^\top + \boldsymbol{W}^e \boldsymbol{e}_i \boldsymbol{e}_k^\top + P_i \boldsymbol{e}_k^\top + P_k \boldsymbol{e}_i^\top] +$$
$$\sum_{j=1}^{l^{(x)}} \frac{1}{\sqrt{d}} \alpha_{ij} \boldsymbol{v} \boldsymbol{e}_j^\top, \qquad (68)$$

where $l^{(x)}$ is the instance length and $\delta_{jk} = 1$ if $j \equiv k$, 0 otherwise. $P_k$ is the positional embedding at position $k$.

We assume that the parameters are initialized as $\boldsymbol{W}_{ij}^e \sim \mathcal{N}(0, \sigma_e^2)$, and $\boldsymbol{v}_j \sim \mathcal{N}(0, \sigma_v^2)$, and the distribution of the attention weights are independent of the parameters $\boldsymbol{W}^e$ and $\boldsymbol{v}$. Let us consider the case where the test input is simply a token $e$. If the network width approaches infinity, $\frac{\boldsymbol{v}^\top \boldsymbol{W}^e \boldsymbol{e}_j}{d} \to 0$ and NTK between the token $e$ and the instance $x$ will converge to $\Theta_\infty(e, x)$, which obeys

$$\Theta_\infty(e, x) \approx \sum_{i=1}^{l^{(x)}} \sum_{j=1}^{l^{(x)}} \mathbb{E}(\alpha_{ij})(\sigma_e^2 + \sigma_v^2) \boldsymbol{e}^\top \boldsymbol{e}_j, \qquad (69)$$

where $\mathbb{E}(\alpha_{ij})$ is the expectation of $\alpha_{ij}$ when elements of $\boldsymbol{W}^e$ obeys $\boldsymbol{W}_{ij}^e \sim \mathcal{N}(0, \sigma_e^2)$.

## A.4 MV Model

The label score of an instance is defined as

$$s = \boldsymbol{v}^\top \sum_j \frac{1}{d\sqrt{d}} \boldsymbol{M}(\boldsymbol{e}_j) \boldsymbol{W}^e \boldsymbol{e}_{j+1}, \qquad (70)$$

where $\boldsymbol{M}(\boldsymbol{e}_j) = \mathrm{diag}(\boldsymbol{W}\boldsymbol{W}^e \boldsymbol{e}_j)$ (diag converts a vector into a diagonal matrix.) and $j = 1, 2, \ldots, l^{(x)} - 1$.

The proof sketch is given as follows: The gradients can be computed as

$$\frac{\partial f}{\partial \boldsymbol{v}} = \frac{1}{d\sqrt{d}} \sum_j (\boldsymbol{W}\boldsymbol{W}^e \boldsymbol{e}_j) \odot (\boldsymbol{W}^e \boldsymbol{e}_{j+1}),$$

$$\frac{\partial f}{\partial \boldsymbol{W}} = \frac{1}{d\sqrt{d}} \sum_j [(\boldsymbol{W}^e \boldsymbol{e}_{j+1}) \odot \boldsymbol{v}](\boldsymbol{W}^e \boldsymbol{e}_j)^\top,$$

$$\frac{\partial f}{\partial \boldsymbol{W}^e} = \frac{1}{d\sqrt{d}} \sum_j \boldsymbol{W}^\top [(\boldsymbol{W}^e \boldsymbol{e}_{j+1}) \odot \boldsymbol{v}] \boldsymbol{e}_j^\top,$$

$$+ \frac{1}{d\sqrt{d}} \sum_j [\boldsymbol{v} \odot (\boldsymbol{W}\boldsymbol{W}^e \boldsymbol{e}_j)] \boldsymbol{e}_{j+1}^\top, \qquad (71)$$

where $\odot$ refers to *element-wise* multiplication.

Given a bigram $e_a e_b$, the NTK will be computed as

$$\theta(e_a e_b, x) = \sum_j$$

$$\frac{\boldsymbol{e}_j^\top}{d^3} (\boldsymbol{W}\boldsymbol{W}^e)^\top \mathrm{diag}(\boldsymbol{W}^e \boldsymbol{e}_{j+1} \odot \boldsymbol{W}^e \boldsymbol{e}_b) \boldsymbol{W}\boldsymbol{W}^e \boldsymbol{e}_a$$

$$+ \frac{\boldsymbol{e}_j^\top}{d^3} (\boldsymbol{W}^e)^\top \boldsymbol{W}^e \boldsymbol{e}_a \boldsymbol{e}_{j+1}^\top (\boldsymbol{W}^e)^\top \mathrm{diag}(\boldsymbol{v})^2 \boldsymbol{W}^e \boldsymbol{e}_b$$

$$+ \frac{\boldsymbol{e}_j^\top}{d^3} \boldsymbol{e}_a \boldsymbol{e}_{j+1}^\top (\boldsymbol{W}^e)^\top \mathrm{diag}(\boldsymbol{v}) \boldsymbol{W}\boldsymbol{W}^\top \mathrm{diag}(\boldsymbol{v}) \boldsymbol{W}^e \boldsymbol{e}_b$$

$$+ \frac{\boldsymbol{e}_{j+1}^\top}{d^3} \boldsymbol{e}_b \boldsymbol{e}_j^\top (\boldsymbol{W}\boldsymbol{W}^e)^\top \mathrm{diag}(\boldsymbol{v})^2 \boldsymbol{W}\boldsymbol{W}^e \boldsymbol{e}_a$$

$$+ \frac{\boldsymbol{e}_{j+1}^\top}{d^3} \boldsymbol{e}_a X(e_b)^\top Y(e_j)$$

$$+ \frac{\boldsymbol{e}_j^\top}{d^3} \boldsymbol{e}_b X(e_{j+1})^\top Y(e_a), \qquad (72)$$

where

$$X(e_b) = \boldsymbol{W}^\top [(\boldsymbol{W}^e \boldsymbol{e}_b) \odot \boldsymbol{v}],$$
$$Y(e_j) = [\boldsymbol{v} \odot (\boldsymbol{W}\boldsymbol{W}^e \boldsymbol{e}_j)]. \qquad (73)$$

When the network width approaches infinity, the NTK will converge to

$$\Theta_\infty(e_a e_b, x) = \sum_j$$

$$\sigma_e^4 \sigma_w^2 \boldsymbol{e}_j^\top \boldsymbol{e}_a \boldsymbol{e}_{j+1}^\top \boldsymbol{e}_b$$
$$+ \sigma_v^2 \sigma_e^2 \sigma_w^2 \boldsymbol{e}_j^\top \boldsymbol{e}_a \boldsymbol{e}_{j+1}^\top \boldsymbol{e}_b$$
$$+ \sigma_v^2 \sigma_e^2 \sigma_w^2 \boldsymbol{e}_j^\top \boldsymbol{e}_a \boldsymbol{e}_{j+1}^\top \boldsymbol{e}_b$$
$$+ \sigma_v^2 \sigma_e^2 \sigma_w^2 \boldsymbol{e}_j^\top \boldsymbol{e}_a \boldsymbol{e}_{j+1}^\top \boldsymbol{e}_b$$
$$= \sum_j (\sigma_e^4 \sigma_w^2 + 3\sigma_v^2 \sigma_e^2 \sigma_w^2) \boldsymbol{e}_j^\top \boldsymbol{e}_a \boldsymbol{e}_{j+1}^\top \boldsymbol{e}_b, \qquad (74)$$

which can capture the co-occurrence between bigrams.

## A.5 L-RNN Model

We follow the work of Emami et al. (2021) and Gu et al. (2021) and focus on a linear RNN, whose hidden state is defined as follows,

$$\boldsymbol{h}_t = \frac{1}{\sqrt{d}}\boldsymbol{W}^h\boldsymbol{h}_{t-1} + \frac{1}{d}\boldsymbol{W}\boldsymbol{W}^e\boldsymbol{e}_t, \quad (75)$$

where $\boldsymbol{W}^h \in \mathbb{R}^{d\times d}$ and $\boldsymbol{W} \in \mathbb{R}^{d\times d}$ and the initial hidden state is a zero vector. We can expand the hidden states across time steps and obtain

$$\boldsymbol{h}_t = \frac{1}{d}\sum_{j=1}^{t}(\frac{\boldsymbol{W}^h}{\sqrt{d}})^{t-j}\boldsymbol{W}\boldsymbol{W}^e\boldsymbol{e}_j, \quad (76)$$

where $(\frac{\boldsymbol{W}^h}{\sqrt{d}})^0 = \boldsymbol{I}$.

The label score of an instance is computed based on the final hidden state as

$$
\begin{aligned}
s &= \boldsymbol{v}^\top\boldsymbol{h}_T \\
&= \frac{1}{d}\sum_{j=1}^{T}\boldsymbol{v}^\top(\frac{\boldsymbol{W}^h}{\sqrt{d}})^{T-j}\boldsymbol{W}\boldsymbol{W}^e\boldsymbol{e}_j \\
&= \sum_{j=1}^{T}s_j,
\end{aligned}
\quad (77)
$$

where $s_j = \frac{1}{d}\boldsymbol{v}^\top(\frac{\boldsymbol{W}^h}{\sqrt{d}})^{T-j}\boldsymbol{W}\boldsymbol{W}^e\boldsymbol{e}_j$ and $T$ is the final time step. Note that $T-j$ means the distance between the current token and the last token in an instance and $s_j$ can be viewed as the label score for the token with a distance of $T-j$ from the last token. The gradients will be calculated as

$$\frac{\partial s}{\partial \boldsymbol{v}} = \frac{1}{d}\sum_{j=1}^{T}(\frac{\boldsymbol{W}^h}{\sqrt{d}})^{T-j}\boldsymbol{W}\boldsymbol{W}^e\boldsymbol{e}_j,$$

$$\frac{\partial s}{\partial \boldsymbol{W}^e} = \frac{1}{d}\sum_{j=1}^{T}[\boldsymbol{v}^\top(\frac{\boldsymbol{W}^h}{\sqrt{d}})^{T-j}\boldsymbol{W}]^\top\boldsymbol{e}_j^\top,$$

$$\frac{\partial s}{\partial \boldsymbol{W}} = \frac{1}{d}\sum_{j=1}^{T}[\boldsymbol{v}^\top(\frac{\boldsymbol{W}^h}{\sqrt{d}})^{T-j}]^\top[\boldsymbol{W}^e\boldsymbol{e}_j]^\top,$$

$$\frac{\partial s}{\partial \boldsymbol{W}^h} =$$

$$\frac{1}{d}\sum_{j=1}^{T}\sum_{k=0}^{T-j-1}[\frac{\boldsymbol{v}^\top}{\sqrt{d}}(\frac{\boldsymbol{W}^h}{\sqrt{d}})^{T-j-k-1}]^\top$$

$$[(\frac{\boldsymbol{W}^h}{\sqrt{d}})^k\boldsymbol{W}\boldsymbol{W}^e\boldsymbol{e}_j]^\top,$$

$$\quad (78)$$

where $T > 1$. When $T = 1$, $\frac{\partial s}{\partial \boldsymbol{W}^h}$ does not exist.

**Lemma A.3.** Assume we initialize parameters following Gaussian distributions, i.e., $\boldsymbol{W}_{ij} \sim \mathcal{N}(0, \sigma_w^2)$, $\boldsymbol{W}_{ij}^h \sim \mathcal{N}(0, \sigma_h^2)$, $\boldsymbol{W}_{ij}^e \sim \mathcal{N}(0, \sigma_e^2)$, and $\boldsymbol{v}_j \sim \mathcal{N}(0, \sigma_v^2)$. When the network width approaches infinity, the NTK $\Theta(x', x)$ converges to a deterministic one $\Theta_\infty(x', x)$, which obeys

$$
\begin{aligned}
\Theta_\infty(x', x) &= \sum_{k=0}^{\min(T',T)-1}\sigma_h^{2k}\sigma_e^2\sigma_w^2\boldsymbol{e}_{T'-k}^\top\boldsymbol{e}_{T-k} \\
&+ \sum_{k=0}^{\min(T',T)-1}\sigma_h^{2k}\sigma_v^2\sigma_w^2\boldsymbol{e}_{T'-k}^\top\boldsymbol{e}_{T-k} \\
&+ \sum_{k=0}^{\min(T',T)-1}\sigma_h^{2k}\sigma_v^2\sigma_e^2\boldsymbol{e}_{T'-k}^\top\boldsymbol{e}_{T-k} \\
&+ \sum_{k=1}^{\min(T',T)-1}k\sigma_h^{2k-2}\sigma_v^2\sigma_w^2\sigma_e^2\boldsymbol{e}_{T'-k}^\top\boldsymbol{e}_{T-k}.
\end{aligned}
$$

where $k = 0, 1, \ldots, \min(T', T) - 1$, which indicates the distance from the last tokens.

*Proof.* The multiplication between $\boldsymbol{W}^h$ and its transpose can be computed as

$$(\frac{\boldsymbol{W}^h}{\sqrt{d}})^\top\frac{\boldsymbol{W}^h}{\sqrt{d}} = \frac{1}{d}(\boldsymbol{W}^h)^\top\boldsymbol{W}^h, \quad (79)$$

where each element $w_{ij}$ in $\boldsymbol{W}^h$ follows $w_{ij} \sim \mathcal{N}(0, \sigma_h^2)$. Each element $w'$ in the output of the multiplication will be

$$
\begin{aligned}
\lim_{d\to\infty}w'_{ij} &= \lim_{d\to\infty}\frac{1}{d}\sum_{r=1}^{d}w_{ri}w_{rj} = \mathbb{E}[w_{ri}w_{rj}] \\
&= \begin{cases}\sigma_h^2, & \text{if } i \equiv j \\ 0, & \text{otherwise}\end{cases}
\end{aligned}
\quad (80)
$$

Hence,

$$\lim_{d\to\infty}(\frac{\boldsymbol{W}^h}{\sqrt{d}})^\top\frac{\boldsymbol{W}^h}{\sqrt{d}} = \sigma_h^2\boldsymbol{I}, \quad (81)$$

where $\boldsymbol{I} \in \mathbb{R}^{d\times d}$ is an identity matrix.

We can also obtain

$$\lim_{d\to\infty}\frac{1}{d}\boldsymbol{v}_\alpha^\top(\frac{\boldsymbol{W}^h}{\sqrt{d}})^k\boldsymbol{v}_\beta = 0, \quad (82)$$

where $k > 0$ is an integer and the elements in vectors $\boldsymbol{v}_\alpha$ and $\boldsymbol{v}_\beta \in \mathbb{R}^d$ are Gaussian distributed with zero means. Based on this, we can arrive at

$$
\begin{aligned}
\lim_{d\to\infty}\frac{1}{d}\boldsymbol{v}_\alpha^\top\left[(\frac{\boldsymbol{W}^h}{\sqrt{d}})^{k'}\right]^\top(\frac{\boldsymbol{W}^h}{\sqrt{d}})^k\boldsymbol{v}_\beta = \\
\sigma_h^{2k}\lim_{d\to\infty}\frac{1}{d}\boldsymbol{v}_\alpha^\top(\frac{\boldsymbol{W}^h}{\sqrt{d}})^{k'-k}\boldsymbol{v}_\beta = 0.
\end{aligned}
\quad (83)
$$

where $k' > k$ (both are integers). A similar conclusion can be obtained for the case $k' < k$.

Given instances $x'$ and $x$, whose label scores are $s'$ and $s$ respectively, we can have

$$\lim_{d \to \infty} < \frac{\partial s'}{\partial \boldsymbol{v}}, \frac{\partial s}{\partial \boldsymbol{v}} >$$
$$= \lim_{d \to \infty} \frac{1}{d^2} \sum_{i=1}^{T'} \sum_{j=1}^{T} \boldsymbol{e}_i^\top (\boldsymbol{W}^e)^\top \boldsymbol{W}^\top \left[ (\frac{\boldsymbol{W}^h}{\sqrt{d}})^{T'-i} \right]^\top$$
$$(\frac{\boldsymbol{W}^h}{\sqrt{d}})^{T-j} \boldsymbol{W} \boldsymbol{W}^e \boldsymbol{e}_j$$
(84)

Like what we have done previously, we can obtain that the elements in $\boldsymbol{e}_i^\top (\boldsymbol{W}^e)^\top \boldsymbol{W}^\top$ and $\boldsymbol{W} \boldsymbol{W}^e \boldsymbol{e}_j$ are Gaussian distributed. Let $\boldsymbol{v}_\alpha^\top = \frac{1}{\sqrt{d}} \boldsymbol{e}_i^\top (\boldsymbol{W}^e)^\top \boldsymbol{W}^\top$ and $\boldsymbol{v}_\beta = \frac{1}{\sqrt{d}} \boldsymbol{W} \boldsymbol{W}^e \boldsymbol{e}_j$, we will get

$$\lim_{d \to \infty} < \frac{\partial s'}{\partial \boldsymbol{v}}, \frac{\partial s}{\partial \boldsymbol{v}} >$$
$$= \sum_{(i,j), T'-i \equiv T-j} \sigma_h^{2k} \sigma_e^2 \sigma_w^2 \boldsymbol{e}_i^\top \boldsymbol{e}_j$$
(85)
$$= \sum_{k=0}^{\min(T',T)-1} \sigma_h^{2k} \sigma_e^2 \sigma_w^2 \boldsymbol{e}_{T'-k}^\top \boldsymbol{e}_{T-k}$$

where $T' - i = T' - j = k$.

Similarly, we can obtain terms $< \frac{\partial s}{\partial \boldsymbol{W}^e}, \frac{\partial s}{\partial \boldsymbol{W}^e} >$, $< \frac{\partial s'}{\partial \boldsymbol{W}}, \frac{\partial s}{\partial \boldsymbol{W}} >$ and $< \frac{\partial s'}{\partial \boldsymbol{W}^h}, \frac{\partial s}{\partial \boldsymbol{W}^h} >$.  □

Let $k = T - j$, we can re-write $s_j$ in Equation 77 as

$$s_{[k]}^e = \frac{1}{d} \boldsymbol{v}^\top (\frac{\boldsymbol{W}^h}{\sqrt{d}})^k \boldsymbol{W} \boldsymbol{W}^e \boldsymbol{e}, \qquad (86)$$

which means the label score for token $e$ at position $k$ from the last token. We thereby define an NTK $\Theta(e, k, x)$ to represent the interaction between token $e$ at position $k$ and instance $x$.

When the network width approaches infinity, the NTK $\Theta(e, k, x)$ converges to a deterministic one $\Theta_\infty(e, k, x)$, which obeys

$$\Theta_\infty(e, k, x) = \rho(k) \boldsymbol{e}^\top \boldsymbol{e}_{T-k}, \qquad (87)$$

where $\rho(k) = \sigma_h^{2k} \sigma_e^2 \sigma_w^2 + \sigma_h^{2k} \sigma_v^2 \sigma_w^2 + \sigma_h^{2k} \sigma_v^2 \sigma_e^2 + k \sigma_h^{2k-2} \sigma_v^2 \sigma_w^2 \sigma_e^2$ and $k$ is a non-negative integer. To make it consistent, we can also replace the instance length with $l^{(x)}$.

### A.6 Multi-class Classification

Compared to the binary architecture in the main paper, the last linear layer will be modified to project the hidden states into a $L$-dimension vector ($L$ is the label space), and the sigmoid layer will be replaced by a *softmax* layer.

The label score of an instance will be described as:

$$\boldsymbol{s}(t) = \boldsymbol{f}_t(x; \theta_t), \qquad (88)$$

where $\boldsymbol{s}(t)$ is a vector with a dimension of $L$. For brevity, we omit the denotation $t$.

We can get the probability distribution for all the labels:

$$\boldsymbol{p} = \text{softmax}(\boldsymbol{s}), \qquad (89)$$

where $\boldsymbol{p} \in R^L$.

The cross-entropy loss will be used, and the loss can be computed as:

$$\mathcal{L} = -\frac{1}{m} \sum_{(x,y) \in \mathcal{D}} \log \left( \boldsymbol{p}^{(x)} \right)^\top \boldsymbol{y}^{(x)}, \qquad (90)$$

where $\boldsymbol{p}^{(x)}$ refers to the *softmax* output for instance $x$ and $\boldsymbol{y}^{(x)}$ is the *one-hot* label vector for instance $x$. The derivative of $\mathcal{L}$ with respect to $vs$ is computed as follows,

$$\frac{\partial \mathcal{L}}{\partial \boldsymbol{s}^{(x)}} = \frac{1}{m} (\boldsymbol{y}^{(x)} - \boldsymbol{p}^{(x)}). \qquad (91)$$

Given a test input $x'$, the learning dynamics of its output from the model with the infinite-width network can be described as

$$\dot{\boldsymbol{s}}' = \frac{1}{m} \sum_{(x,y) \in \mathcal{D}} (\boldsymbol{y}^{(x)} - \boldsymbol{p}^{(x)}) \Theta_\infty(x', x), \qquad (92)$$

where $\Theta_\infty(x', x) \in \mathbb{R}^{L \times L}$ refers to the converged NTK determined at initialization.

We can have a similar analysis on this dynamics as $(\boldsymbol{y}^{(x)} - \boldsymbol{p}^{(x)})$ will only be positive in the dimension where $\boldsymbol{y}_z^{(x)} = 1$ ($z$ is the dimension index in $\boldsymbol{y}$). Suppose $\Theta_\infty(x', x) \in \mathbb{R}^{L \times L}$ works in a way that increases the influence of the corresponding dimension $z$ in $\dot{\boldsymbol{s}}'$ when $x'$ is associated with label $z$, i.e., the label score corresponding to such a label will receive a positive gain and grow to be large.

## B  Influence of Activation Functions

As we previously mentioned, the MLP model can have an induced bias for the features learned between positive tokens and negative tokens, while the self-attention model does not have this problem. Actually, this problem is caused by the activation function *ReLU* used in the MLP model. Apart from the *ReLU* function, we considered the *identity map* ($\phi = \boldsymbol{I}$), and *tanh* activation functions.

Let us consider token $e$ from the vocabulary and its NTK with the training instance $x$.

$\phi = \boldsymbol{I}$  In this case, the model is linear. When the network width approaches infinity, $\Theta(x', x)$ converges to a deterministic NTK $\Theta_\infty(x', x)$ during training which obeys

$$
\begin{aligned}
\Theta_\infty(x', x) =& \sum_{i=1}^{l^{(x')}} \sum_{j=1}^{l^{(x)}} \sigma_e^2 \sigma_w^2 \boldsymbol{e}_i^\top \boldsymbol{e}_j \\
&+ \sum_{i=1}^{l^{(x')}} \sum_{j=1}^{l^{(x)}} \sigma_e^2 \sigma_v^2 \boldsymbol{e}_i^\top \boldsymbol{e}_j \\
&+ \sum_{i=1}^{l^{(x')}} \sum_{j=1}^{l^{(x)}} \sigma_w^2 \sigma_v^2 \boldsymbol{e}_i^\top \boldsymbol{e}_j \\
=& \sum_{i=1}^{l^{(x')}} \sum_{j=1}^{l^{(x)}} (\sigma_e^2 \sigma_w^2 + \sigma_e^2 \sigma_v^2 + \sigma_w^2 \sigma_v^2) \boldsymbol{e}_i^\top \boldsymbol{e}_j.
\end{aligned}
\tag{93}
$$

Given token $e$, the NTK $\Theta_\infty(e, x)$ obeys

$$
\Theta_\infty(e, x) = \sum_{j=1}^{l^{(x)}} (\sigma_e^2 \sigma_w^2 + \sigma_e^2 \sigma_v^2 + \sigma_w^2 \sigma_v^2) \boldsymbol{e}^\top \boldsymbol{e}_j,
\tag{94}
$$

which means the NTK is affected by the frequency that $e$ is seen in instance $x$. This is similar to the MLP model with the *ReLU* activation function but without the induced bias.

$\phi = \tanh$  Suppose $\phi = \tanh$, then we have that

$$
\begin{aligned}
&\lim_{d \to \infty} \frac{1}{d} \boldsymbol{\phi}^\top(\boldsymbol{c}_e) \boldsymbol{\phi}(\boldsymbol{c}_j) \\
&= \lim_{d \to \infty} \frac{1}{d} \tanh^\top(\boldsymbol{c}_e) \tanh \phi(\boldsymbol{c}_j) \\
&= \mathbb{E}[\tanh(\boldsymbol{c}_{er}) \tanh(\boldsymbol{c}_{jr})],
\end{aligned}
\tag{95}
$$

where $r$ is the row index and $\mathbb{E}[\tanh(\boldsymbol{c}_{er}) \tanh(\boldsymbol{c}_{jr})]$ is a constant regardless of $r$.

It can be inferred that $\boldsymbol{c}_{er}$ yields to a Gaussian distribution with a zero mean. As $\tanh$ is an odd function, we will arrive at $\mathbb{E}[\tanh(\boldsymbol{c}_{er})] = 0$.

Suppose $\boldsymbol{e}^\top \boldsymbol{e}_j = 0$, namely, $e \neq e_j$, $\tanh(\boldsymbol{c}_{er})$ and $\tanh(\boldsymbol{c}_{jr})$ are two independent random variables and $\mathbb{E}[\tanh(\boldsymbol{c}_{er}) \tanh(\boldsymbol{c}_{jr})] = 0$. Similarly, when $e \neq e_j$, we can obtain

$$
\begin{aligned}
\lim_{d \to \infty} \frac{1}{d^2} \boldsymbol{v}^\top \boldsymbol{D} \boldsymbol{D}_j \boldsymbol{v} \boldsymbol{e}^\top (\boldsymbol{W}^e)^\top \boldsymbol{W}^e \boldsymbol{e}_j = 0 \\
\lim_{d \to \infty} \frac{1}{d^2} \boldsymbol{v}^\top \boldsymbol{D} \boldsymbol{W} (\boldsymbol{W})^\top \boldsymbol{D}_j \boldsymbol{v} \boldsymbol{e}^\top \boldsymbol{e}_j = 0.
\end{aligned}
\tag{96}
$$

| Dataset | | Train | Dev | Test |
|---|---|---|---|---|
| PTB | Token Num | 887,521 | 70,390 | 78,669 |
| | Vocab Size | | 10,000 | |
| Wiki2 | Token Num | 2,088,628 | 217,646 | 245,569 |
| | Vocab Size | | 33,278 | |
| Shakespeare | Token Num | 1,003,854 | 111,540 | - |
| | Vocab Size | | 65 | |

Table 4: Language modeling datasets statistics.

| Dataset | | MLP | CNN | SA | MV | L-RNN |
|---|---|---|---|---|---|---|
| SST | valid | 79.4 | 79.7 | 80.4 | 77.3 | 79.4 |
| | test | 81.5 | 81.0 | 80.7 | 77.8 | 80.7 |
| SSTwsub | valid | 79.6 | 78.7 | 79.9 | 80.5 | 76.9 |
| | test | 79.0 | 79.2 | 80.3 | 81.3 | 77.5 |
| IMDB | valid | 91.5 | 91.8 | 91.7 | - | - |
| | test | 91.8 | 91.4 | 91.6 | - | - |
| Agnews | valid | 91.5 | 91.4 | 91.7 | - | - |
| | test | 91.3 | 91.1 | 91.5 | - | - |

Table 5: Accuracy (%) on the datasets. Adagrad optimizers. "-" refers to that the training is unstable.

Suppose $\boldsymbol{e}^\top \boldsymbol{e}_j = 1$, we can obtain

$$
\begin{aligned}
&\mathbb{E}[\tanh(\boldsymbol{c}_{er}) \tanh(\boldsymbol{c}_{jr})] = \mathbb{E}[\tanh^2(\boldsymbol{c}_{er})] \\
&\lim_{d \to \infty} \frac{1}{d^2} \boldsymbol{v}^\top \boldsymbol{D} \boldsymbol{D}_j \boldsymbol{v} \boldsymbol{e}^\top (\boldsymbol{W}^e)^\top \boldsymbol{W}^e \boldsymbol{e}_j = \\
&\sigma_e^2 \sigma_v^2 \mathbb{E}[\boldsymbol{D}_{jrr}^2] \boldsymbol{e}^\top \boldsymbol{e}_j \\
&\lim_{d \to \infty} \frac{1}{d^2} \boldsymbol{v}^\top \boldsymbol{D} \boldsymbol{W} (\boldsymbol{W})^\top \boldsymbol{D}_j \boldsymbol{v} \boldsymbol{e}^\top \boldsymbol{e}_j = \\
&\sigma_w^2 \sigma_v^2 \mathbb{E}[\boldsymbol{D}_{jrr}^2] \boldsymbol{e}^\top \boldsymbol{e}_j,
\end{aligned}
\tag{97}
$$

where $\boldsymbol{D}_{jrr}$ is the $r$-th diagonal element in $\boldsymbol{D}_j$. Given token $e$, the converged NTK is computed as

$$
\begin{aligned}
\Theta_\infty(e, x) =& \sum_{j=1}^{l^{(x)}} \mathbb{E}[\tanh^2(\boldsymbol{c}_{er})] \boldsymbol{e}^\top \boldsymbol{e}_j + \\
&\sum_{j=1}^{l^{(x)}} (\sigma_e^2 \sigma_v^2 \mathbb{E}[\boldsymbol{D}_{jrr}^2] + \sigma_w^2 \sigma_v^2 \mathbb{E}[\boldsymbol{D}_{jrr}^2]) \boldsymbol{e}^\top \boldsymbol{e}_j.
\end{aligned}
\tag{98}
$$

This indicates the MLP model with the $\tanh$ activation does not have the induced bias.

## C  More Experimental Results

The statistics of language modeling datasets are listed in Table 4. The performances are listed in Table 5[10].

**Extracted Tokens&Bigrams**  We automatically extracted tokens associated with specific labels as shown in Table 6.

---

[10]The MV and L-RNN models are not stable during training on IMDB and Agnews.

| Dataset | Extracted Token (bigram) Num |
|---------|------------------------------|
| SST | 412 (+)/ 265 (-) |
| SSTwsub | 73 (+)/ 47 (-) |
| SSTwsub (bigram) | 779 (+)/ 542 (-) |
| IMDB | 414 (+)/363 (-) |
| Agnews | 441 (I)/540 (II)/297 (III)/334 (IV) |

Table 6: Numbers of extracted tokens. "(+)" and "(-)" refer to tokens associated with the *positive* and *negative* labels, respectively. "(I)", "(II)", "(III)", and "(IV)" refer to the Class 1-4 of Agnews.

Adjectives with polarity were extracted automatically from SSTwsub. We first extracted tokens that were seen more frequently in either positive or negative instances than in the other instances, i.e., the frequency ratio either larger than 3 or less than 1/3. Then we used the *textblob* package [11] to find out adjectives from those extracted tokens. Examples are shown in Table 7.

| | Tokens |
|---|--------|
| + | inventive, nice, authentic, sympathetic, lovable, grand, happy, enthusiastic, noble, detailed, exotic, remarkable, charismatic, ... |
| - | inexplicable, feeble, sloppy, disastrous, stupid, terrible, unhappy, horrible, atrocious, idiotic, angry, uninspired, vicious, unfocused, artificial, ... |

Table 7: Examples of the extracted positive adjectives ("+") and negative adjectives ("-") from the SSTwsub dataset.

## C.1 Language Modeling

Language modeling can also be viewed as a classification task, where the label space is the vocabulary size and each label is a token in the vocabulary. The model output before the *softmax* layer is a vector with a dimension of the vocabulary size. Each dimension corresponds to a label, i.e., a token in the vocabulary. We extracted 20 most frequent words from the PTB [12] and Wiki2 datasets (Merity et al., 2016), respectively. For each extracted token, we searched for the top 30 token-label pairs. In addition, we searched for the bottom 30 token-label pairs for comparison. We trained a two-layer Transformer language model[13] on PTB and Wiki2, with the embeddings size and hidden size of 200. Figure 7 shows that the co-occurrence can be generally captured by the Transformer model. Specifically, given an extracted token, its output (with a dimension of the vocabulary size) will likely have

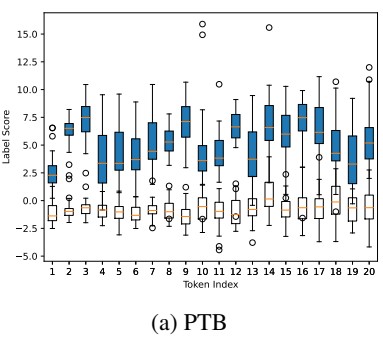

(a) PTB

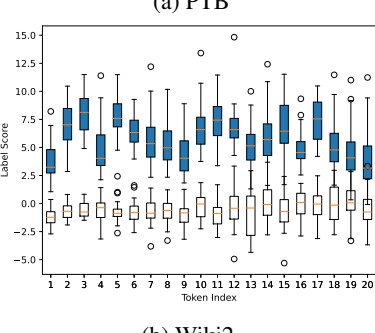

(b) Wiki2

Figure 7: Distributions of the label scores for majorly co-occurring token pairs (in blue) and rarely co-occurring token pairs.

relatively larger values in the positions corresponding to the majorly co-occurring tokens than in the positions corresponding to the rarely co-occurring tokens.

For the experiment on nanoGPT, we followed the settings in the quick start part and trained the model on the char-level Shakespeare dataset. The max iteration number is set as 2000 instead of the default 5000. We extracted the most 20 frequent chars and calculated their co-occurrences with each char in the vocabulary (65 chars in total). For each extracted char, we extracted the top 5 and bottom 5 co-occurring chars, respectively from the vocabulary. Feeding each extracted char into the model, we could get the output vector, as well as the label scores for majorly co-occurring chars and rarely co-occurring chars.

## C.2 Negation

Aside from the results on the self-attention model, we also conducted experiments to verify the ability to capture negation phenomena on the CNN and the L-RNN models. It can be seen from Figure 8 that both the CNN and Transformer models can capture the negation for the positive adjectives. But it seems they do not capture such phenomena on the negative adjectives. The CNN model handles the negation bigrams in a linear combination way, and the polarity of a negation bigram is the combination of the polarity of the two tokens involved. As the

[11] https://textblob.readthedocs.io/en/dev/
[12] https://catalog.ldc.upenn.edu/docs/LDC95T7/cl93.html
[13] https://pytorch.org/tutorials/beginner/transformer_tutorial.html

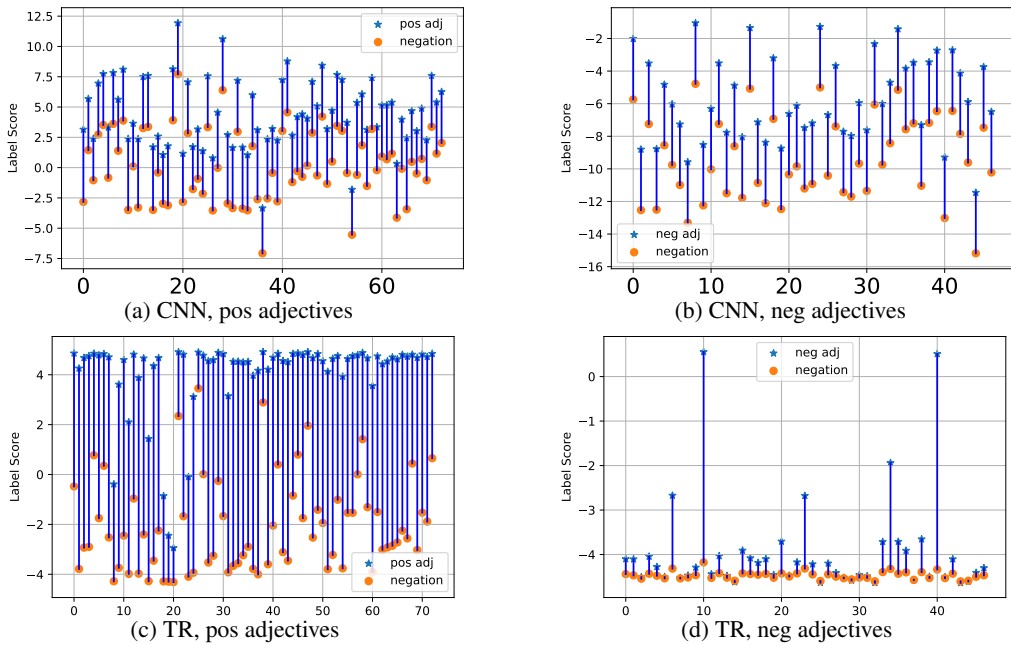

Figure 8: Label scores for the *positive adjectives* (pos adjectives) and *negative adjectives* (neg adjectives) as well as their negation expressions. "TR" refers to the Transformer model (one head, one layer)

.

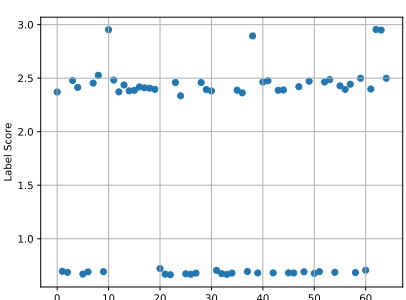

Figure 9: The label score differences between the subphrases and the phrases. SSTwsub. SA model.

Figure 10: Label scores for extracted positive tokens and negative tokens from SST. MLP with *SiLU*.

token *not* appears more in negative instances, it is assigned negative label scores. Therefore, adding the token *not* to a positive adjective can weaken its positive polarity but for a negative adjective, adding *not* can strengthen its negative polarity. This also implies the limitation of such models: they rely largely on token-label features.

We also extracted 65 phrases (less than 11 words, 17 positive and 48 negative) starting with negation words *not*, *never*, and *hardly* from SSTwsub. We computed their label scores and the label scores of their sub-phrase constructed by removing the negation words. Figure 9 shows the differences between the label scores from the subphrases and the phrases are all positive, indicating the negation words play negative roles in a linear combination and do not reverse the polarity of negative subphrases.

### C.3 Feature Extraction

**SiLU** Figure 10 shows *SiLU* can also prevent an induced bias in the features captured.

**Adam Optimizer** We conduct experiments on the Adam optimizer and observed patterns (shown in Figure 11) similar to those from the Adagrad optimizer in the main paper.

**L-RNN** Extracting sufficient tokens that appear in a specific position and a specific category of instances in real-world datasets like SST is not easy. Instead, we created a synthetic dataset (1,000 positive instances and 1,000 negative instances) based on three types of tokens. One type of token is seen in positive instances with a fixed distance from the last tokens; another type of token is seen in negative instances with a fixed distance from the last tokens.

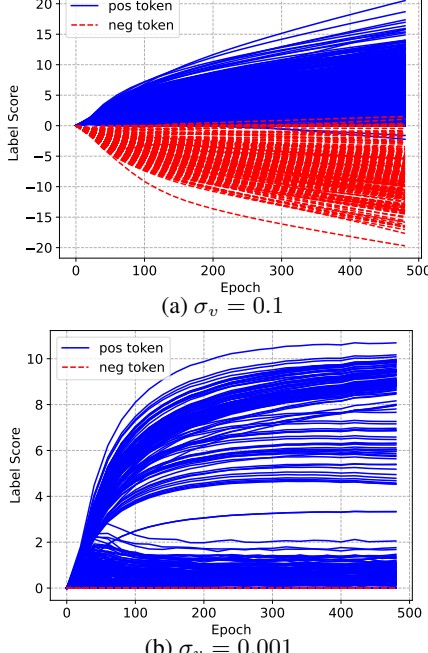

(a) $\sigma_v = 0.1$

(b) $\sigma_v = 0.001$

Figure 11: Label scores for the extracted tokens from SST. Adam optimizer.

The other tokens are seen randomly in both positive and negative instances with random positions. In this experiment, we set the fixed distance from the last ones as 2. Adagrad optimizers were used. It can be seen from Figure 12 that when $k = 2$, the positive and negative tokens are assigned significant label scores, while when $k = 0$ and $k = 4$, the label scores are less significant, supporting our aforementioned analysis on the L-RNN model.

**IMDB**  IMDB is a dataset with relatively long instances. Our findings can also be observed on the IMDB dataset in Figure 13. Particularly, we could also observe a feature bias on the IMDB dataset (as shown in Figure 13c) when we used the MLP model with *ReLU*, supporting our analysis in the main paper again that *ReLu* may cause a feature bias. However, we did not observe an obvious performance decline on the IMDB dataset.

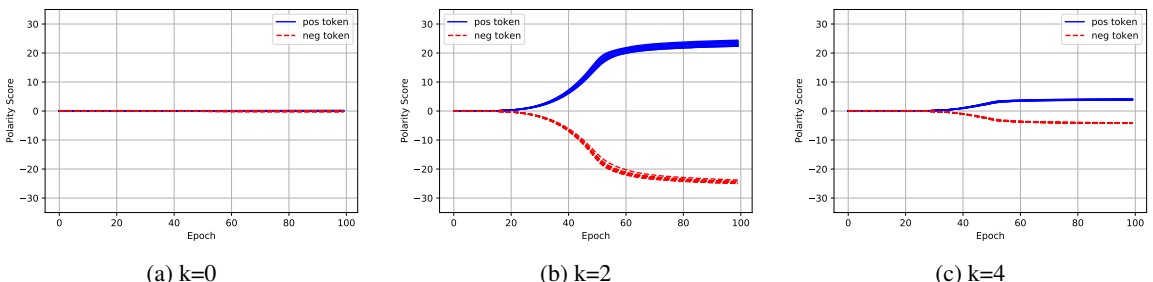

(a) k=0                    (b) k=2                    (c) k=4

Figure 12: Label scores for the *positive tokens* (pos token) and *negative tokens* (neg token) at different positions for the L-RNN model. $k$ refers to the distance from the last tokens. Synthetic dataset.

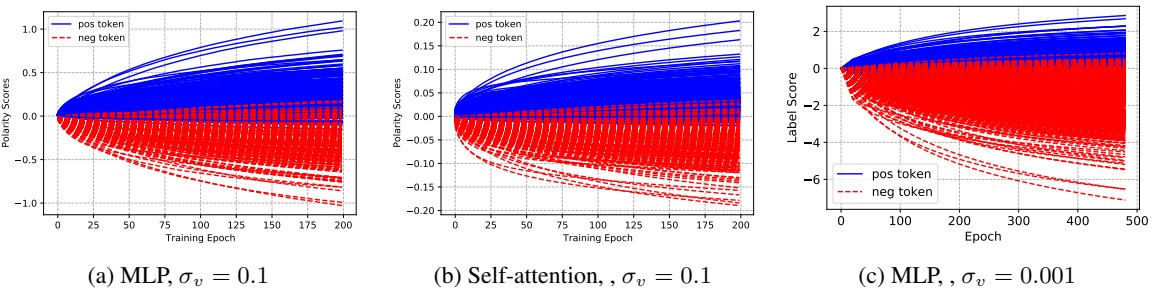

(a) MLP, $\sigma_v = 0.1$          (b) Self-attention, , $\sigma_v = 0.1$          (c) MLP, , $\sigma_v = 0.001$

Figure 13: Label scores for extracted tokens from IMDB. $d = 64$.