# OpenReview forum: "Unraveling Feature Extraction Mechanisms in Neural Networks"
_EMNLP/2023/Conference — EMNLP 2023 Main_

### Official Review · Reviewer_ak69 · 2023-07-27

**Soundness:** 5

**Excitement:**

4: Strong: This paper deepens the understanding of some phenomenon or lowers the barriers to an existing research direction.

**Paper Topic And Main Contributions:**


The paper describes a novel theoretical perspective on feature extraction mechanisms for fundamental neural modules when using gradient descent in text classification tasks. The authors begin with a theory around the learning dynamics for binary text classifiers. Following, the authors validate the proposed work with experimentation focusing on a deeper understanding of the factors that affect feature extraction and limitations of the models.


**Reasons To Accept:**

- The theory is presented well and covers both binary classification and multi-class classification (appendix).
- The experiment results delve deep into some interesting behavioral aspects of neural network modules setting the stage for future work. For example, the authors describe the decline in the performance of MLP with ReLU.
- Learning dynamics of popular neural network modules such as MLP, CNN, and SA have been explained. This sets the stage for more complex modules.


**Reasons To Reject:**

- The results, experiments, and some theories described in the main paper require the reader to refer to the appendix often. As an example, theory and experiments around MV, and L-RNN are described in the appendix rather than the main paper.
- This paper is better suited for a longer journal paper rather than a full conference paper. That being said, the authors can figure out if they want to just present a subset of the work for this conference.
- It is a little unclear to what degree the learning dynamics change without the infinite width condition.


**Reproducibility:**

4: Could mostly reproduce the results, but there may be some variation because of sample variance or minor variations in their interpretation of the protocol or method.

**Reviewer Confidence:**

3: Pretty sure, but there's a chance I missed something. Although I have a good feel for this area in general, I did not carefully check the paper's details, e.g., the math, experimental design, or novelty.

---

> ### Author Rebuttal · Authors · 2023-08-29
>
> Thanks very much for your valuable comments and suggestions!
> Due to the strict page limit, we had to prioritize the main content in the body of the paper and present the detailed theories and additional experiments in the appendix. However, we recognize the importance of ensuring a smoother reading experience and will consider restructuring the content for better flow.
>
> The infinite width condition serves as a mathematical tool to elucidate the learning dynamics more clearly. Prior work, such as that by Sanjeev Arora et al. (2019), delved into the nuances of the infinite width condition. Their findings suggest that when the network width grows sufficiently large, the Neural Tangent Kernel (NTK) can approximate the infinite width NTK effectively while ensuring the error remains within specific bounds. In our empirical studies, we observed that the outcomes from finite network widths mirrored the theoretical insights derived from the infinite width assumption closely. To provide greater clarity on this topic and address potential ambiguities, we will expand on the discussions on the influence of finite and infinite widths in our revised version.

---

### Official Review · Reviewer_tfVt · 2023-07-29

**Soundness:** 4

**Excitement:**

4: Strong: This paper deepens the understanding of some phenomenon or lowers the barriers to an existing research direction.

**Paper Topic And Main Contributions:**

The problem that this paper aims to address is to obtain more understanding of what type of features are learned by neural networks of fundamental models on tokenized data. The paper studies the neural tangent kernel (NTK) at the infinite-width limit for fundamental models (MLP, CNN, self-attention, matrix-vector, and linear RNN models).

The contributions and findings are several. The learning dynamics of particular tokens' scores depend on a term that is characterized by the co-occurrence of the token with the training data, and another term that does not depend on co-occurrences, and thus represents a bias in the features. Different fundamental models have bias, but notably, some do not, such as self-attention. The choice of variance of the initialization of the final layer, and the choice of activation functions affect the types of features learned. The theory also suggests the reason why ReLU might be replaced with other activations in Transformers, and why increasing the dataset size matters (because of the increased diversity of co-occurrences).

The theory is in the infinite-width but in practice, for finite width, the trends observed in the theory hold.

**Questions For The Authors:**

Having a bias is typically quite useful if the inductive bias is chosen well. Can your theory suggest what kind of biases might be useful? Can we start from engineering a term in the learning dynamics and then reverse-engineer an architecture that will induce that bias?

**Reasons To Accept:**

1. Well-written paper with clean motivation and intuitive explanation of the derivations.

2. Theoretical study the trends of which reflect empirical studies.

3. Could be a useful framework to study novel phenomena in training fundamental models.

**Reasons To Reject:**

1. We do not see a phenomenon that has not been observed empirically to be predicted by the theory. Typically when introducing a theory we hope to find insight that could help us build better feature extractors. Could you comment on that or show some promise for future work?

**Reproducibility:**

4: Could mostly reproduce the results, but there may be some variation because of sample variance or minor variations in their interpretation of the protocol or method.

**Reviewer Confidence:**

3: Pretty sure, but there's a chance I missed something. Although I have a good feel for this area in general, I did not carefully check the paper's details, e.g., the math, experimental design, or novelty.

**Typos Grammar Style And Presentation Improvements:**

Figure 1 is not referenced early enough and it is not very intuitive/ descriptive. Please improve the caption of the figure.

---

> ### Author Rebuttal · Authors · 2023-08-29
>
> Thanks very much for your constructive feedback!
>
> 1. Promise for future work:
> Our main intention was to provide a theoretical grounding that mirrors the observed learning dynamics across various neural architectures. In our paper, we indeed discussed the potential limitations of the target models in their feature extraction capabilities. As an illustration, our observations confirmed that the one-layer self-attention model did not effectively capture the negation phenomenon, especially concerning negative adjectives. Contrastingly, the matrix-vector model demonstrated superior efficacy in this domain, which suggests that multiplication-based models might inherently be better at n-gram feature extraction (as discussed in Lines 605-609). Given these insights, there's a significant prospect in developing matrix-vector multiplication models tailored for downstream tasks with a strong reliance on n-gram features.
>
>
> 2. Discussion about biases:
> So far as we know, an inductive bias pertains to the inherent assumptions within a model that shape its predictions for data that it has not encountered before. In our work, the inductive bias posits that both positive and negative tokens are anticipated to impartially contribute to the ultimate decision-making process. Therefore, we believe the biases fostering this objective are useful.
> In the context of our work, the bias exhibited by the MLP with ReLU specifically refers to the second term in Equation 8. A crucial observation we made was that such biases could compromise the fairness of the knowledge that the model extracts. As corroborated by our results in Table 2, the influence of this bias could, in certain scenarios, detrimentally affect performance, underscoring the importance of appropriately chosen biases.
> While our current work sheds light on these dynamics, it may offer insights into the roles of biases induced by architecture during gradient descent and help us tune their influences based on their roles. For example, the bias exhibited by the MLP with ReLU may potentially hinder classification performance. We may consider reverse-engineering an architecture devoid of this activation function.

---

### Official Review · Reviewer_SPeX · 2023-08-05

**Soundness:** 4

**Excitement:**

4: Strong: This paper deepens the understanding of some phenomenon or lowers the barriers to an existing research direction.

**Paper Topic And Main Contributions:**

The goal of the paper is to understand the learning dynamics of a neural network model. Specifically, the authors propose a theoretical approach to analyze the feature extraction mechanism of neural networks of infinite width using Neural Tangent Kernels. They analyze multi-layer perception, CNNs, linear RNNs, self-attention mechanism, and matrix-vector model and come up with theoretical models to analyze the output of these models. They train models with various activation functions and initialization on SST, Agnews, and IMDB datasets. The findings of their experiments indicate that reducing the initial variance of the final layer's weight can result in a substantial feature bias. This bias makes positive tokens more significant than negative ones in both the MLP and CNN models. The experiments also demonstrate that alternatives to ReLU such as GeLU and SiLU are more robust to changes in initialization compared to ReLU.They demonstrate that co-occurring characters in a given context have a higher model output score compared to non-co-occurring characters. They hypothesize that this may be a part of the reason why large datasets, which provide a large amount of co-occurrence information, helps improve the performance of deep learning models.

**Reasons To Accept:**

* The authors provide theoretical models to analyze the learning mechanism in MLPs, CNNs, and self-attention layers.
* Empirical results provide insightful results relating to activation layers and the sensitivity of models to initialization.
* Future work in this direction, especially on larger neural networks, can benefit from the groundwork laid in this paper.

**Reasons To Reject:**

* The empirical results mostly offer a correlational analysis and not a causal or mechanistic understanding of the underlying phenomenon.

**Reproducibility:**

4: Could mostly reproduce the results, but there may be some variation because of sample variance or minor variations in their interpretation of the protocol or method.

**Reviewer Confidence:**

2: Willing to defend my evaluation, but it is fairly likely that I missed some details, didn't understand some central points, or can't be sure about the novelty of the work.

**Typos Grammar Style And Presentation Improvements:**

Figure 3 is too small. Consider increasing the size of the plot or at least increasing the font size of the x and y axis labels.

---

> ### Author Rebuttal · Authors · 2023-08-29
>
> Thanks very much for your valuable comments and suggestions!
>
> We appreciate the importance of differentiating between correlational and causal analyses. The primary intention of our paper was not to present a causal analysis. Nevertheless, our work offers some causal insights where we considered the influence of significant factors in the learning dynamics and developed experiments to verify the phenomena predicted.
> For example, as shown in our manuscript, Equation 8 indicates the presence of a bias in the evolution of the token label score for the MLP model with ReLU activation. This bias can cause an imbalance between positive tokens and negative tokens. Furthermore, we have expanded on this discussion in section A.1.3 (in the appendix), emphasizing the role of variance σ_v in influencing this bias. As shown in Figures 2a and 2d, with the change of σ_v, the bias becomes pronounced, thereby diminishing the prominence of negative tokens. We believe these discussions indicate a form of causal analysis.
>
> In addition, we would like to clarify that we did not seek to conduct correlational analysis in our experiments. The main intention is to validate our analysis in terms of the phenomena predicted (e.g., Figure 2) based on the terms in the learning dynamics, not investigating the statistical correlation between factors and output.
>
> We acknowledge the feedback regarding the potential ambiguity in our presentation. We will revise our paper to clarify our primary intention and explain in sections where the causal insights are discussed. We will also adjust Figure 3 for better readability.

---

### Meta-Review · Area_Chair_xvry · 2023-09-19

**Recommendation:** 5

**Metareview:**

This paper is a solid work and very interesting. By making NTK, the work studies the Feature Extraction Mechanisms of existing models, such as self-attention, cnn, etc.

---

### Decision · Program_Chairs · 2023-10-07

**Decision:**

Accept-Main

**Comment:**

This paper is a solid work and very interesting. By making NTK, the work studies the Feature Extraction Mechanisms of existing models, such as self-attention, cnn, etc.